# Impact of constitutional *TET2* haploinsufficiency on molecular and clinical phenotype in humans

Eevi Kaasinen et al.[#]

Clonal hematopoiesis driven by somatic heterozygous *TET2* loss is linked to malignant degeneration via consequent aberrant DNA methylation, and possibly to cardiovascular disease via increased cytokine and chemokine expression as reported in mice. Here, we discover a germline *TET2* mutation in a lymphoma family. We observe neither unusual predisposition to atherosclerosis nor abnormal pro-inflammatory cytokine or chemokine expression. The latter finding is confirmed in cells from three additional unrelated *TET2* germline mutation carriers. The *TET2* defect elevates blood DNA methylation levels, especially at active enhancers and cell-type specific regulatory regions with binding sequences of master transcription factors involved in hematopoiesis. The regions display reduced methylation relative to all open chromatin regions in four *DNMT3A* germline mutation carriers, potentially due to TET2-mediated oxidation. Our findings provide insight into the interplay between epigenetic modulators and transcription factor activity in hematological neoplasia, but do not confirm the putative role of TET2 in atherosclerosis.

Correspondence and requests for materials should be addressed to O.K. (email: outi.kilpivaara@helsinki.fi) or to L.A.A. (email: lauri.aaltonen@helsinki.fi) or (email: lauri.aaltonen@ki.se). [#]A full list of authors and their affiliations appears at the end of the paper.

Clonal hematopoiesis (CH) is common in aged individuals and bears implications to health through risk of malignant degeneration of cells[1] and possible risk of cardiovascular disease (CVD)[2–4]. Heterozygous tet methylcytosine dioxygenase 2 (*TET2*) and DNA methyltransferase 3A (*DNMT3A*) mutations are the two most frequent drivers of CH[1], and mechanistic studies connecting CH and CVD revolve around *Tet2*-deficient models. Heterozygous and homozygous *Tet2* loss in mice accelerates atherosclerosis, possibly via enhanced macrophage-driven inflammation[4,5]. Acceleration of heart failure has also been suggested[6]. In particular, two macrophage-mediated mechanisms have been proposed: exacerbated expression and inflammasome-mediated secretion of interleukin (IL)-1β, as well as aberrant chemokine expression signature[4,5]. These findings have promoted hopes for population level prevention of CVD through detection of individuals with *TET2*-mutation-positive clones and subsequent use of cholesterol-lowering medications or compounds targeting IL-1β and other inflammatory pathways[2,4,5]. Thus, understanding whether heterozygous *TET2* loss is associated with CVD in humans, and if yes through what mechanism, is of utmost importance.

DNA methylation is a key regulator of cell development and differentiation, and its aberrations are an essential factor in hematological neoplasia[7]. DNA methylation is mediated by DNA methyltransferase enzymes that transfer a methyl group to carbon atom 5 of cytosine nucleotide at CpG dinucleotides or CxG context at gene bodies, x standing for bases T, A, or C[8]. In DNA demethylation, TET protein family of dioxygenases catalyze the oxidization of 5-methylcytosine (5-mC) to 5-hydroxymethylcytosine (5-hmC), 5-formylcytosine, and 5-carboxylcytosine, acting as an initiator of DNA demethylation cascade subsequently resulting in an unmodified cytosine[8].

In addition to CH, somatic frameshift, nonsense, and missense *TET2* mutations are commonly seen for example in myelodysplastic syndrome (6–26% prevalence), acute myeloid leukemia (AML; 12–27% in adult de novo AML), chronic myelomonocytic leukemia (20–58%), and angioimmunoblastic T-cell lymphoma (33–83%)[8]. Although a key event, TET2 loss alone is not sufficient to trigger malignancy[7]. Careful examination of individuals with a germline mutation could provide valuable insight into the effects of TET2 loss in humans. In this study, we observed the effects of constitutional heterozygous *TET2* loss in a unique pedigree of seven carriers segregating a truncating *TET2* germline mutation, as well as one case of de novo *TET2* germline mutation. For these individuals, extensive clinical documentation was available. Methylation analysis of four individuals with a *DNMT3A* germline mutation as well as analysis of inflammatory response in two additional *TET2* germline mutation carriers reported earlier by Schaub et al.[9] provided further context to the results.

## Results

**Study subjects**. The Finnish family segregating a *TET2* germline mutation is presented in Fig. 1a. Ly1 was diagnosed with nodular lymphocyte predominant Hodgkin lymphoma (NLPHL) at age 46 (Supplementary Table 1, Supplementary Fig. 1), and Ly2 at age 45. At age 52, Ly2 experienced a relapse diagnosed as T-cell-rich B-cell lymphoma. Ly3 was diagnosed with NLPHL at age 39. Relapse at age 41 was diagnosed as mixed-cellular Hodgkin lymphoma. Clinical bone marrow examination was done twice (with 7 years' time interval) for Ly1 and Ly2 after lymphoma diagnosis. As the only finding of note, Ly2 had slightly hyperplastic bone marrow in the second examination. Whole-genome (Ly1) and exome (Ly2 and Ly3) sequencing analysis revealed a heterozygous one-base deletion NM_001127208.2:c.4500delA in

*TET2* (Fig. 1b, Supplementary Table 2). The mother of Ly1, Ly2, and Ly3 was found to be a carrier of the deletion based on analysis of archival tissue DNA, and three further carriers (Ly9, Ly11, and Ly14) were found in the next generations. The deletion is similar to those often seen in somatic form in hematological neoplasia and causes a frameshift at lysine 1500 residue, resulting in a premature stop codon 70 residues later (NP_001120680.1:p.Lys1500AsnfsTer71) (Fig. 1c). An AML patient has been found with the same mutation previously[10]. Multiple amino-acid residues critical for the structural integrity of TET2[11] are lost due to the deletion (Fig. 1d–f), and although both alleles are expressed at the mRNA level (Supplementary Fig. 2), the truncation leads to heterozygous loss of TET2 at the protein level (Fig. 1g, Supplementary Fig. 3). *TET2* c.4500delA (hereafter TET2delA) was absent in 5197 in-house controls, the Exome Aggregation Consortium dataset (version 0.2; http://exac.broadinstitute.org/), and in 80,000 individuals in the Genome Aggregation Database (version 2.0; http://gnomad.broadinstitute.org/).

Through Northern Finland Intellectual Disability (NFID) Cohort, we could identify a carrier of de novo nonsense mutation in *TET2* (NM_001127208.2:c.1471C>T, NP_001120680.1:p.Gln491Ter; hereafter individual Id1 with TET2X mutation). His developmental milestones had been normal except for linguistic skills, and he received speech therapy until age 7. Comprehensive etiological examinations including copy number analysis did not reveal the cause of developmental delay[12]. He was diagnosed with mild mental retardation at age 7 and had features of attention deficit hyperactivity disorder. According to the latest examination by Wechsler Intelligence Scale For Children–IV at age 16, he had reached normal cognitive skills apart from verbal comprehension. His older sister had experienced a delay in speech development but managed in standard school education. The parents were confirmed as TET2X-negative by Sanger sequencing, and the patient's germline status was confirmed from saliva-derived DNA.

To provide context to molecular findings in the *TET2* mutation carriers, DNA from non-carrier relatives, unrelated controls, as well as from four *DNMT3A* germline mutation carriers from NFID Cohort displaying features of Tatton-Brown-Rahman syndrome[13] were available for methylation analyses (Supplementary Table 3). In addition, we had the opportunity to study immune responses in blood monocytes isolated from two carriers of a heterozygous 4 base pair (bp) deletion in *TET2* (hereafter TET2del4) reported earlier by Schaub et al.[9] (Fig. 1c).

**Blood DNA methylation**. Since deficient DNA demethylation is the key consequence of *TET2* mutations in hematological neoplasia[14,15], we compared blood DNA methylation by targeted and whole-genome bisulfite sequencing (WGBS) in *TET2* mutation carriers and non-carriers (Supplementary Fig. 4, Supplementary Tables 3, and 4). For comparisons between TET2delA carriers and mutation-free controls, targeted data from 5 carriers and 10 controls were utilized, as the higher number of samples reduces the likelihood that changes are caused by disproportions in cell types between the two groups. WGBS data, created from the three lymphoma-free TET2delA carriers and three age-matched controls, were used for genomic enrichment analyses due to the higher number of data points and because the chromatin annotation of affected regions after TET2 loss is likely identical in different cell types.

TET2delA carriers displayed a significantly higher degree of overall hypermethylation (two-sided Wilcoxon rank sum test, *p*-value 0.003) and decreased hypomethylation (two-sided Wilcoxon rank sum test, *p*-value 0.01), compatible with demethylation deficiency (Fig. 2a, Supplementary Table 5).

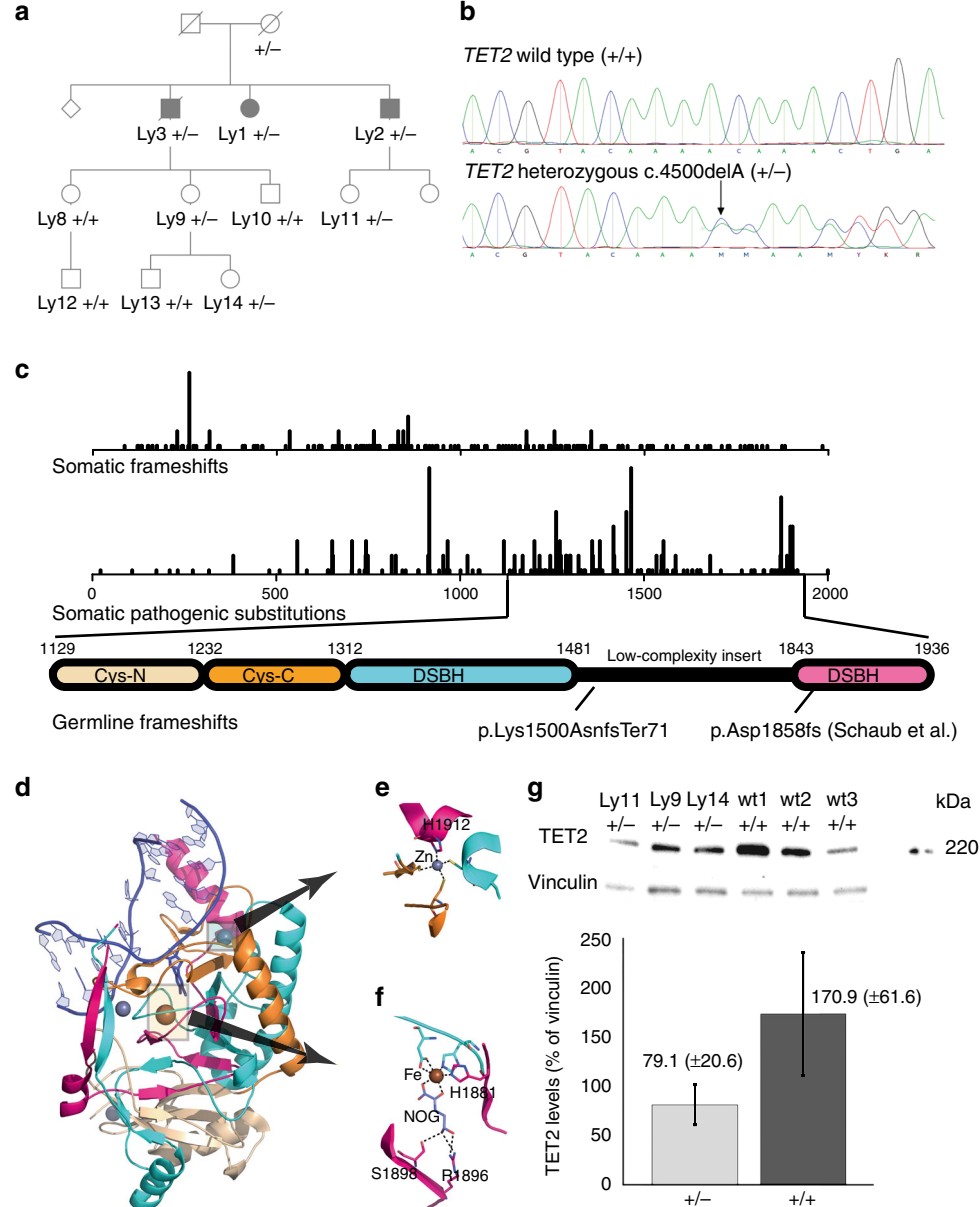

**Fig. 1** *TET2* c.4500delA causes loss of the protein. **a** Pedigree (details modified for confidentiality) showing *TET2* c.4500delA (TET2delA) statuses; mutation carriers are marked with $+/-$ and wild-type individuals with $+/+$. Solid square (male) or circle (female) depicts patients with lymphoma. **b** Electropherograms of TET2delA-mutated and wild-type sample. **c** (Upper section) Somatic frameshifts or pathogenic substitutions along *TET2* in hematopoietic and lymphoid tissue malignancies from COSMIC database (release v81). (Lower section) Protein domain structure of TET2 from Hu et al.[11]. The C-terminal catalytic domain of TET2 comprises Cys-rich N-terminal (Cys-N) and C-terminal (Cys-C) subdomains, and a double-stranded β helix (DSBH) domain. Locations of germline frameshift mutations identified in this study and previously (Schaub et al.[9]) are shown. **d–f** Cartoon representation of the catalytic domain of human TET2 complexed with methylated DNA (PDB entry 4nm6). Overall structure of the wild-type protein (**d**). DNA is colored with blue, flanking Cys-rich domains are colored with light and dark orange, the DSBH core of the protein is colored with cyan and pink, the latter depicting the sequence lost due to the TET2delA mutation. The predicted loss includes residues involved in formation of the DSBH domain such as H1912 involved in coordination of Zn3-ion (**e**), as well as residues R1896 and S1898 critical for binding the substrate *N*-oxalyglycine (NOG) (**f**), H1881 involved in coordination of Fe-ion (**f**), and Y1902 and H1904 interacting with 5-methylcytosine. Proper folding of the mutant protein lacking these critical amino acids appears highly unlikely based on the structure. *N*-oxalyglycine, 5-methylcytosine, and residues mentioned above are shown with stick representation. An iron and three zinc ions are shown as brown and gray balls, respectively. **g** Representative western blot of TET2 from lymphoblastoid cells of three TET2delA mutation carriers and three wild-type individuals of the family. The graph shows the quantification of TET2 protein normalized to the vinculin. Bars represent the mean ± s.d. of TET2 intensities

Altogether 644/660 (97.6%) of differentially methylated regions (DMRs) between TET2delA carriers and controls were hypermethylated (Supplementary Data 1). To confirm that similar changes are not found in carriers of common *TET2* single-nucleotide polymorphisms (SNPs), we compared carriers of heterozygous TET2 p.Leu1721Trp and p.Pro363Leu ($n = 4$; rs34402524 and rs17253672 found in the same haplotype) to non-carriers ($n = 11$), derived from two Finnish families. Blood DNA methylation of all 15 individuals was measured by targeted bisulfite sequencing similar to TET2delA carriers and

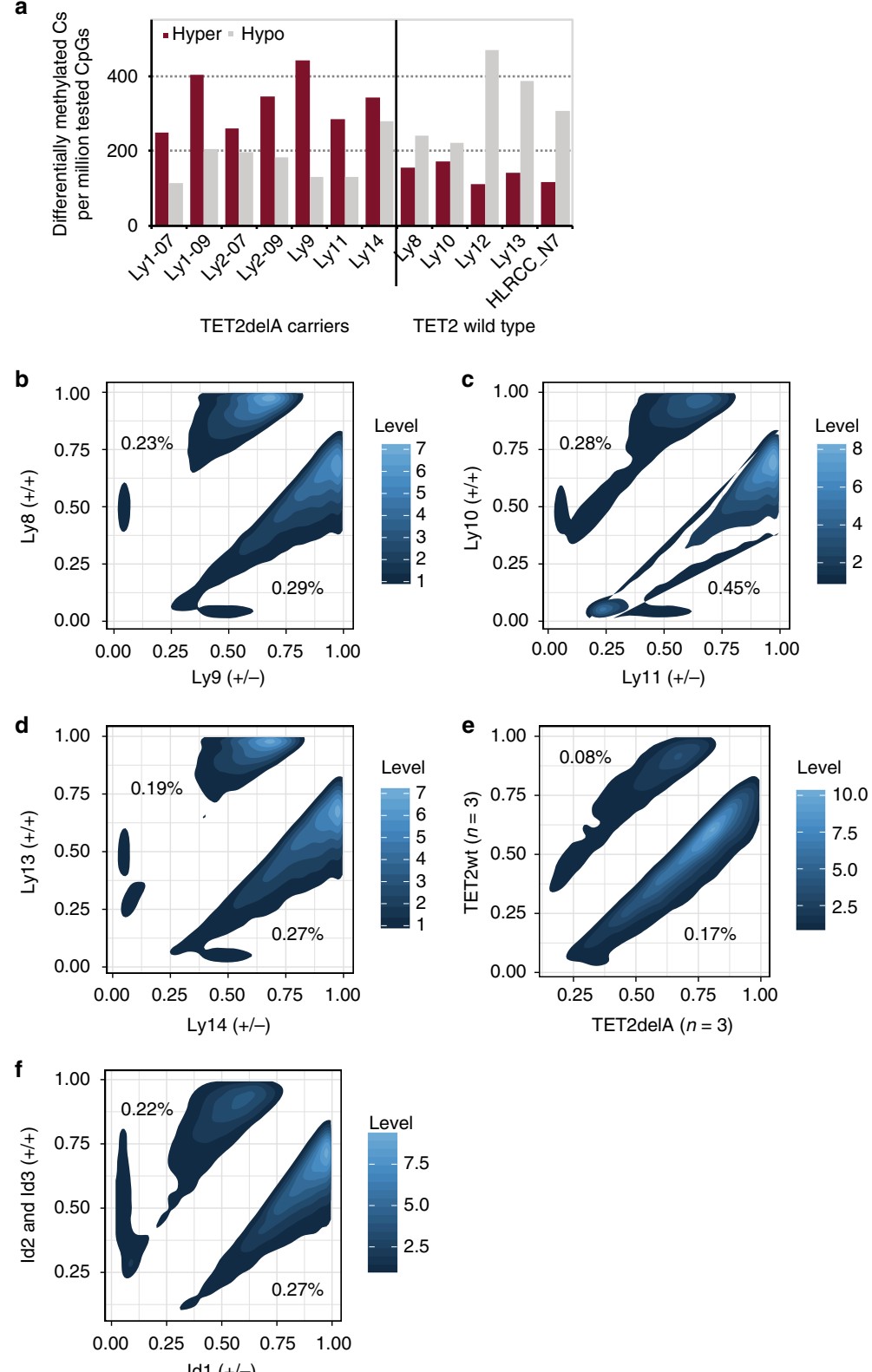

controls. No difference in the number of hypermethylated (two-sided Wilcoxon rank sum test, *p*-value 0.5714) nor hypomethylated (two-sided Wilcoxon rank sum test, *p*-value 0.1773) CpG sites was observed between the groups. Lymphoma patients (Ly1 and Ly2) and cancer-free TET2delA carriers (Ly9, Ly11, and Ly14) were also compared and no skewing toward

hypermethylation was observed, although the results should be interpreted with caution due to low number of individuals in this comparison (Supplementary Data 2). The demethylation deficiency caused by heterozygous TET2 loss was further confirmed by WGBS. Hypermethylation was more prominent than hypomethylation when WGBS data of three cancer-free

**Fig. 2** Heterozygous *TET2* loss causes increased DNA methylation. **a** Count of hypermethylated (dark red bars) and hypomethylated (light gray bars) cytosines per million tested CpGs in TET2delA carriers (on the left side) and wild-type controls (on the right side). Mutation carriers display elevated hypermethylation levels in blood as compared to non-carriers. Each sample was compared to the set of five baseline control samples in order to derive counts of hyper- and hypomethylated CpGs. Methylation profiling of the patients Ly1 and Ly2 was performed from two separate DNA samples extracted in years 2007 and 2009, others were sampled once. **b–f** Density plots of differentially methylated CpGs from whole-genome bisulfite sequencing (WGBS). **b–d** Pairwise comparison of each TET2delA carrier and age-matched wild-type family member. **e** Group-wise comparison of three TET2delA carriers to three age-matched non-carriers of the family. **f** Pairwise comparison of TET2X carrier and age-matched wild-type individuals from the Northern Finland Intellectual Disability Cohort. In **b–f**, X- and Y-axis represent the methylation fraction in *TET2* mutation carriers and non-carriers, respectively

TET2delA carriers and TET2X carrier were compared to age-matched non-carriers (Fig. 2b–f). While repetitive regions of the genome cannot be evaluated with short reads, the detected hypermethylated CpG sites located mostly at CpG islands of the genome (Supplementary Fig. 5a).

Next, we examined whether the hypermethylation in TET2X and TET2delA carriers was enriched in regions with regulatory function. We analyzed the overlap of the differentially methylated CpG (DMC) sites from the WGBS data with respect to regions displaying known chromatin marks in various primary human blood cell types derived from the Roadmap Epigenomics Project. Hypermethylation showed strongest enrichment at normally H3K4me1-marked chromatin across the different blood cell types (Fig. 3a). Individual comparisons of TET2X and TET2delA mutation carriers to their age-matched controls showed similar enrichment (Supplementary Fig. 6-9). The regions with enriched hypermethylation at H3K4me1 displayed H3K27ac but not H3K27me3 chromatin mark in the same blood cell types, implying that methylation was increased in normally active enhancer regions (Supplementary Fig. 5b). Hypomethylation, on the other hand, showed strongest enrichment at normally H3K9me3-marked, transcriptionally repressed chromatin in all primary human blood cells characterized in the Roadmap Epigenomics Project (Fig. 3b).

As the enhancer regions with increased methylation are likely to have lower activity, we studied chromatin immunoprecipitation followed by high-throughput sequencing (ChIP-seq) of H3K27ac from immortalized lymphoblastoid cell lines. ChIP-seq enrichment analysis revealed reduced histone acetylation at CpG islands as a function of methylation increase in TET2delA carriers as compared to non-carriers (Pearson's correlation; $r = -0.06$, 95% confidence interval $-0.08$ to $-0.03$, and $p$-value $2.192 \times 10^{-06}$) (Fig. 3c).

Overall DNA methylation level of the hypomethylated CpGs in partially methylated domains (PMDs) has been proposed to represent a biomarker for cellular aging[16]. We extracted methylation values from the CpGs in the PMD regions and did not observe methylation difference between the three cancer-free TET2delA carriers and their age-matched non-carrier family members in the WGBS data (Supplementary Fig. 10).

We further compared enrichment of hyper- and hypomethylation at the chromatin marks in blood DNA samples collected at 10 years' time intervals, available from two carriers (Ly9 and Ly11) and two non-carriers (Ly8 and Ly10). Age-related methylation changes showed a similar pattern of enrichment across the different chromatin marks between TET2delA and wild-type individuals (Supplementary Fig. 11).

We next examined whether methylation changes in TET2delA carriers had occurred in hematopoietic cell lineage-specific open chromatin regions that we derived from public single-cell ATAC-seq data[17]. We could observe significantly increased methylation in the targeted bisulfite sequencing data of TET2delA carriers as compared to 10 controls in some of the lineage-specific regions, in particular those found in monocytes, granulocyte/macrophage progenitors, common lymphoid progenitors, B cells, and

megakaryocyte/erythroid progenitors (two-sided Wilcoxon rank sum test, false discovery rate (FDR) < 0.05) (Fig. 4a). Because the emergence of hematopoietic lineages depends on activity of transcription factors (TFs) such as PU.1, RUNX1/2, CEBPα/β, GATA1/2, PAX5, and TBX21[18], we scrutinized the methylation statuses of the consensus binding sequences of these TFs within the lineage-specific open chromatin regions (Supplementary Fig. 12). We found that many cell-type-specific open chromatin regions at RUNX2 (binding specificity similar to RUNX1 and RUNX3), GATA1 (binding specificity similar to GATA2 and GATA3), and PU.1 binding sequences were significantly enriched for methylation in the TET2delA carriers (two-tailed permutation $p$-values corrected for multiple testing via FDR < 0.05) (Fig. 4b, Supplementary Table 6).

To scrutinize further the methylation changes related to gene defects commonly detected in CH and hematological malignancies[1], and to provide additional context to findings in *TET2* mutation carriers, we derived targeted bisulfite sequencing data from blood DNA of four *DNMT3A* germline mutation carriers and their age-matched controls from the NFID Cohort (Supplementary Table 3). *DNMT3A* mutation carriers displayed increased overall hypomethylation (Fig. 4c) and decreased methylation at open chromatin regions of all blood cell types (Supplementary Fig. 13a). Comparison of the effects of TET2 and DNMT3A loss in mouse hematopoietic stem cells has been reported[19]. By enrichment analysis of DMC sites from our WGBS data, we could observe that hypermethylation in *TET2* mutation carriers was most prominent in regions reported to be sites of DNMT3A and TET2 competition[19] (Supplementary Fig. 13b). Thus, we next examined whether the methylation changes caused by human *DNMT3A* germline mutation could be seen in the same regulatory regions as in TET2delA carriers. Indeed, the TF-binding sites at lineage-specific open chromatin regions affected in the TET2delA carriers were affected also in *DNMT3A* mutation carriers, albeit to the opposite direction, showing decreased methylation (Pearson's correlation of relative ratios; $r = 0.31$, 95% confidence interval 0.03–0.54, and $p$-value 0.03) (Fig. 4d).

**Blood immune cell phenotyping and single-cell analysis**. Leukocyte counts of TET2X and TET2delA carriers were grossly normal (Supplementary Table 7), whereas further T- and B-lymphocyte subtyping analyses revealed marked changes (Supplementary Table 8). The two tested TET2delA carriers previously treated for lymphoma (Supplementary Table 1) displayed impaired B-cell maturation, compatible with chemotherapy. The young, approximately 20 years of age TET2delA and TET2X carriers (Ly14 and Id1) had increased percentages of recent thymic emigrant and naive subsets of CD3+CD4+ T cells. The most notable overall changes in the *TET2* mutation carriers were the high percentages of activated (CD38lowCD21low) CD19+ B cells and low percentages of CD3+CD4/8+CCR7−CD45RA− T effector memory cells. In addition, the elderly mutation carriers (Ly1, Ly2, Ly9, and Ly11, aged 36–64 years) exhibited skewing of CD3+CD8+ T cells toward increased proportions of the

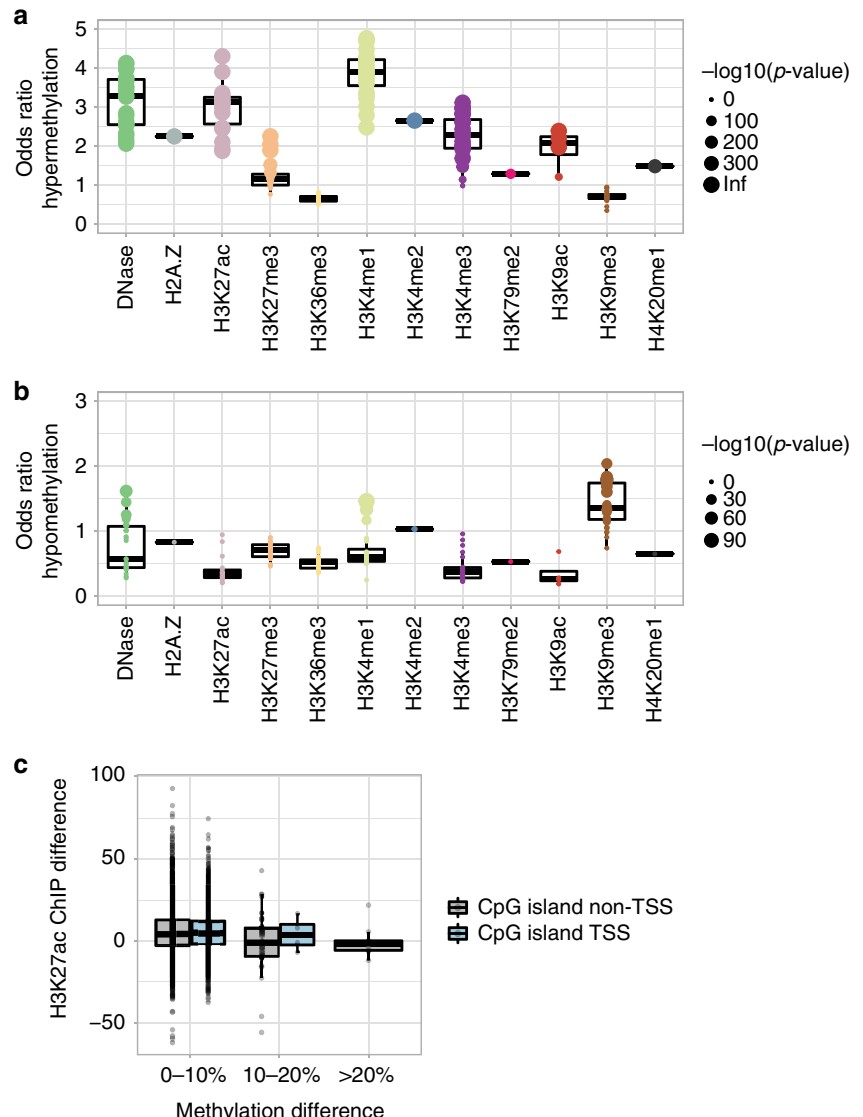

**Fig. 3** Hypermethylation caused by TET2 loss is enriched at active enhancer regions. **a, b** Regions with different chromatin marks in 24 different subtypes of primary human blood cells (each subtype represented by a dot) were available from the Roadmap Epigenomics project. These were compared to the hyper- and hypomethylated sites detected in blood whole-genome bisulfite sequencing (WGBS) of three cancer-free TET2delA carriers relative to three age-matched non-carriers. **a** The hypermethylated sites in TET2delA carriers were characterized by frequent presence of normally H3K4me1-marked chromatin in primary blood cells. In addition, hypermethylation was enriched strongly at DNaseI hypersensitive and H3K27ac-marked chromatin, suggesting that methylation was increased in enhancer regions that are normally active. **b** The hypomethylated sites in TET2delA carriers were enriched at normally H3K9me3-marked regions that typically represent transcriptionally repressed heterochromatin. Odds ratios and p-values from the Fisher's exact test implemented in LOLA R package. **c** Chromatin immunoprecipitation with an anti-H3K27ac antibody in lymphoblastoid cells from TET2delA carriers (Ly9, Ly11, and Ly14) was compared to that from wild-type individuals (Ly8 and Ly10). The difference in the enrichment of active H3K27ac-marked chromatin at CpG islands is displayed as a function of increasing methylation at CpG islands in whole blood of the TET2delA carriers as measured by WGBS. Negative correlation is stronger at CpG islands outside transcription start sites (TSS). Boxplots show the median, and the first and third quartiles

CD3+CD8+CCR7−CD45RA+ terminally differentiated effector memory cells re-expressing CD45RA (TEMRA) phenotype, which in the absence of chronic viremia or lymphopenia suggests immune exhaustion.

Next, we performed single-cell RNA-sequencing analysis of peripheral blood of the five living TET2delA carriers as well as three non-carrier family members. The main cell types were identified through cluster-specific expression patterns and were present in similar fractions in both groups although individual variation was observed (Fig. 5a, Supplementary Fig. 14a). Compatible with the most prominent effect of TET2 loss on

methylation in monocytes (Fig. 4a), TET2 expression was highest in this cell type and mutation carriers did not display compensation through increased TET1 or TET3 expression (Fig. 5b, Supplementary Fig. 14b). A slight decrease in DNMT3A expression was observed in CD4+ T cells of TET2delA carriers (Supplementary Fig. 14b). We then scrutinized the differentially expressed genes between cancer-free TET2delA carriers and age-matched wild-type individuals (Supplementary Tables 9-12). The most prominent difference was increased CXCR4 expression, especially in natural killer (NK) and CD4+/CD8+ T cells (Fig. 5c). We also detected significantly elevated expression of

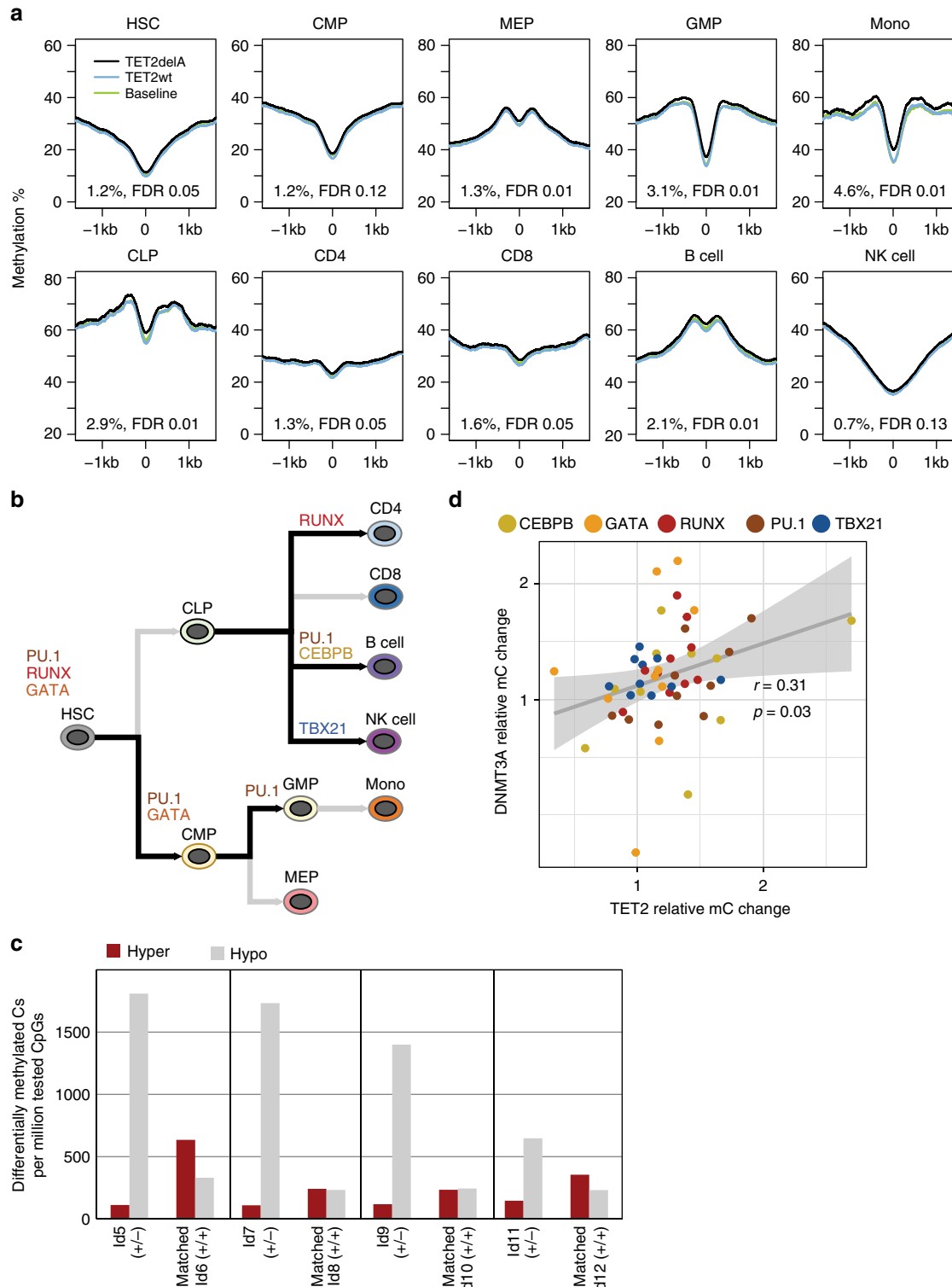

*TSC22D3* in NK cells, CD8+ T cells, and monocytes of cancer-free TET2delA carriers (Fig. 5d). *TSC22D3* is the gene with the highest positive correlation to *CXCR4* expression at whole transcriptome level in public expression data from AML (*n* = 162, The Cancer Genome Atlas data[20] in cBioPortal) (Supplementary Fig. 15). Furthermore, we detected reduced expression of *JCHAIN* (*IgJ*), *IGLL5*, and *AL928768.3*, a human monoclonal IgA1-IgA2λ hybrid molecule from *IgH* locus, especially in B cells of *TET2* mutation carriers (Supplementary Table 11). Compatible

with expression changes of immunoglobulin genes, we detected significantly elevated methylation in a B-cell-specific open chromatin region at the promoter of the *IGHJ6* gene in TET2delA carriers (Supplementary Fig. 16, Supplementary Data 1). Analysis of regulon activity by looking at TFs and their co-expressed *cis* target genes revealed 26 TFs with significantly increased and 12 TFs with significantly decreased activity in cancer-free TET2delA carriers as compared to wild-type individuals, including *TBX21* and *EOMES* that displayed lower activity especially in

**Fig. 4** TET2- and DNMT3A-mediated methylation changes are enhanced at open chromatin with master transcription factor-binding sequences (TFBSs). **a** Methylation is significantly increased in TET2delA carriers at lineage-specific open chromatin regions of monocytes, granulocyte/macrophage progenitors (GMPs), common lymphoid progenitors (CLPs), B cells, and megakaryocyte/erythroid progenitors (MEPs). Percentages represent average methylation differences at the respective open chromatin regions between mutation carriers (Ly1, Ly2, Ly9, Ly11, and Ly14) and 10 controls (Ly8, Ly10, Ly12, Ly13, HLRCC_N7, and controls 1–5) from targeted bisulfite sequencing. False discovery rate (FDR)-adjusted *p*-values are from two-sided Wilcoxon rank sum test. **b** Schematic of the human hematopoietic cell lineage hierarchy showing the 10 cell types analyzed in **a**. Black arrows depict emergence of lineages that showed significant enrichment of methylation at indicated TFBSs at lineage-specific open chromatin regions in the TET2delA carriers. **c** Count of hypermethylated (dark red bars) and hypomethylated (light gray bars) cytosines per million tested CpGs in *DNMT3A* mutation carriers (+/−) and age-matched controls (+/+) from the Northern Finland Intellectual Disability Cohort. Each sample was compared to the same set of five baseline controls as in Fig. 2a. **d** Magnitudes of the methylation changes at master TFBSs located in cell-specific open chromatin regions correlate in *TET2* and *DNMT3A* mutation carriers, albeit these methylation changes occur toward different directions. *X*- and *Y*-axis represent average methylation change in mutation carriers as compared to controls at open chromatin regions with master TFBS relative to all open chromatin regions in each of the 10 cell types. Each dot of a particular color represents one of the 10 cell types, respectively. Pearson's product-moment correlation coefficient (*r*) and *p*-value (*p*) are calculated for the relative ratios. Methylation values from TET2delA carriers were compared to the 10 controls as in **a**, and those from *DNMT3A* mutation carriers to the four controls as in **c**. RUNX and GATA represent binding sequences of RUNX1/2/3 and GATA1/2/3, respectively

---

CD8+ T cells and NK cells (Supplementary Fig. 17, Supplementary Data 3 and 4).

**Clonal hematopoiesis**. The analysis of low-frequency variants from deep exome sequencing revealed CH in two mutation carriers (Ly2 and Ly9) (Supplementary Fig. 18). In Ly1, T-cell receptor gamma rearrangement study had been performed 16 years after lymphoma diagnosis and revealed two clonal PCR products, compatible with CH also in this patient. No somatic *TET2* mutations were detected, suggesting that complete *TET2* loss provided little selective advantage.

**Inflammatory responses of monocytes and macrophages**. Fuster et al.[5] linked the acceleration of atherosclerotic plaque development in *Tet2*-deficient mice to increased NLRP3 inflammasome-mediated secretion of IL-1β by macrophages. To explore this mechanism in human cells, we first primed monocyte-derived macrophages from cancer-free TET2delA carriers (Ly9, Ly11, and Ly14) and controls (Ly8 and an unrelated control) with lipopolysaccharide (LPS) and interferon-γ, followed by stimulation with NLRP3 inflammasome activators. In human macrophages, priming with LPS induces pro-IL-1β and NLRP3 receptor mRNAs, but a second NLRP3-activating stimulus is subsequently required to trigger the caspase-1-mediated proteolytic maturation and secretion of the inflammasome target cytokines, IL-1β and IL-18[21]. TET2delA status did not correlate with elevated secretion of mature IL-1β or IL-18 in response to any of the studied canonical or noncanonical activators of the NLRP3 inflammasome (Fig. 6a, Supplementary Fig. 19a). A trend toward increased levels of activated caspase-1 during NLRP3 inflammasome activation was observed in TET2delA carriers (Fig. 6b). In agreement with cytokine secretion, the mRNA expression of IL-1β, IL-18, and NLRP3 inflammasome complex components was comparable in TET2delA carriers and controls (Fig. 6c, Supplementary Fig. 19b). In mouse macrophages, TET2 represses IL-1β mRNA expression indirectly via trichostatin A (TSA)-sensitive histone deacetylases (HDACs)[5]. We found that addition of TSA during LPS + interferon-γ priming boosted the mRNA induction of IL-1β and NLRP3 receptor both in TET2delA carriers and controls (Fig. 6c), but it did not affect the subsequent caspase-1 activation step (Fig. 6b) or secretion of IL-1β (Supplementary Fig. 19c). To further corroborate these findings and to ensure that the ex vivo monocyte-to-macrophage differentiation process caused no bias, we studied NLRP3 inflammasome responses directly in blood monocytes isolated from carriers of two further heterozygous *TET2*-disrupting mutations, TET2del4 and TET2X (two and one individuals, respectively). Human monocytes (but not

macrophages) express an additional alternative NLRP3 inflammasome pathway triggered by prolonged exposure to soluble LPS[21–23], which was studied alongside the canonical ATP-triggered pathway. We found similar levels of IL-1β secretion in monocytes from *TET2* mutation carriers and controls (Fig. 6d, e). Finally, we silenced *TET2* expression in human peripheral blood monocyte-derived macrophages from six donors using a pool of small interfering RNAs (siRNAs) to mimic heterozygous TET2 loss. The mean *TET2* knockdown was −59 ± 6% and no compensation or siRNA off-target effects were detected in *TET3* expression (Supplementary Fig. 19e), while *TET1* expression was not detected in accordance with previous reports[24]. In agreement with our data from *TET2* germline mutation carriers, *TET2* knockdown had no effect on IL-1β secretion or mRNA induction (Fig. 6g, Supplementary Fig. 19f).

Aberrantly elevated chemokine expression has also been reported in *Tet2*+/− and *Tet2*−/− mouse macrophages[4,5], yet the TET2delA carriers (*n* = 3) did not display increased plasma levels of the key human chemokine IL-8 compared to 12 healthy volunteers (Fig. 6f), and the same was true when CXCL8/IL-8 transcript levels were measured from TET2delA carrier macrophages, as well as after TET2 knockdown by siRNA (Supplementary Fig. 19d and f). Furthermore, the aberrant expression of pro-inflammatory cytokines and chemokines that was reported in the context of *Tet2*+/− mice[4,5] was not detected in humans with RNA-sequencing of TET2delA and control macrophages (Supplementary Data 5). Enrichment analysis of overexpressed genes (*q* < 0.1) identified no significant differences between wild-type and mutant macrophages at baseline nor after LPS + interferon-γ treatment (Supplementary Data 5 and 6). Based on previous works, we selected panels of genes with modified expression during primary human monocyte-to-macrophage differentiation and activation[25,26]. We grouped these genes under cell cycle, macrophage differentiation, and macrophage M1 polarization, the latter induced by LPS + interferon-γ treatment[25]. Clustering of the human macrophage RNA-sequencing samples based on these expression panels found no distinction between the TET2delA carrier and control groups, indicating similar pace of differentiation and cytokine-mediated polarization (Supplementary Fig. 20). Moreover, expression of IL-6 was reported as a target of TET2-HDAC2-mediated transcriptional repression in mouse macrophages and dendritic cells[5,27]. Thus, we further analyzed in our dataset the expression of IL-6 and a panel of eight similarly TET2-regulated LPS-inducible target genes reported in mice[27]. We could not detect consistently higher levels of IL-6 or the co-regulated transcripts in TET2delA carrier macrophages compared to controls (Supplementary Fig. 21) and, in agreement,

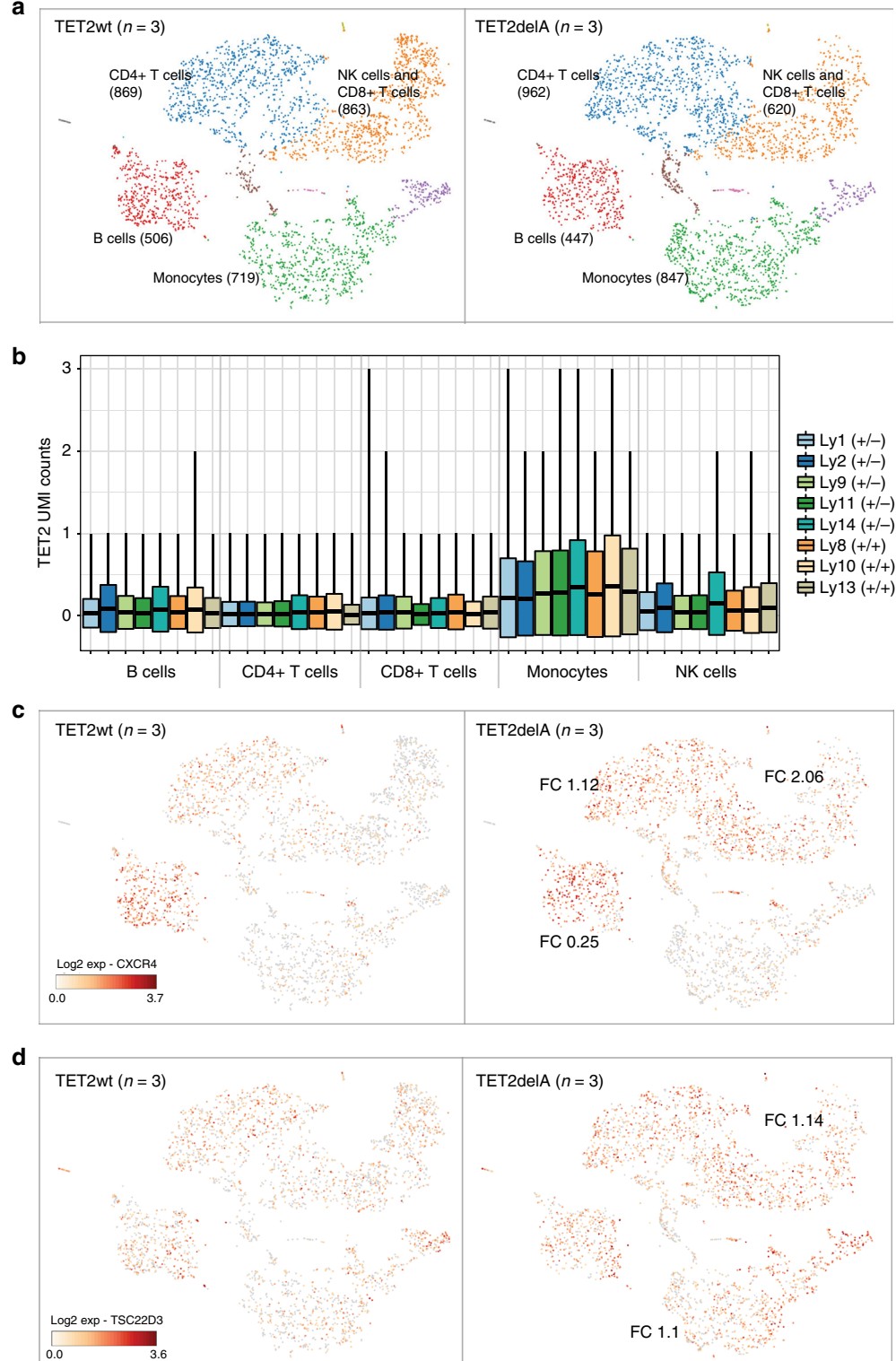

**Fig. 5** Single-cell RNA-sequencing analysis from peripheral blood. **a** The main cell types were present in similar numbers between cancer-free TET2delA carriers (Ly9, Ly11, and Ly14; right) and wild-type individuals (Ly8, Ly10, and Ly13; left). Colors represent blood cells with similar expression profiles in *K*-means (*K* = 10) clustering. Each point represents a cell in the coordinates specified by the two t-SNE (t-distributed stochastic neighbor embedding) components. **b** *TET2* expression in each of identified cell types represented with unique molecular identifier (UMI) counts. Boxplot shows the mean ± standard deviation. TET2delA carriers are marked with +/− and wild-type individuals with +/+. **c** *CXCR4* was the most significantly increased transcript in natural killer (NK) cells and CD8+ T cells [log2 fold change (FC) 2.06], CD4+ T cells (FC 1.12), and all cell types compiled (FC 0.92), in cancer-free mutation carriers (right) as compared to non-carriers (left). **d** *TSC22D3* expression was significantly increased in NK cells and CD8+ T cells (FC 1.14) and monocytes (FC 1.1) of mutation carriers

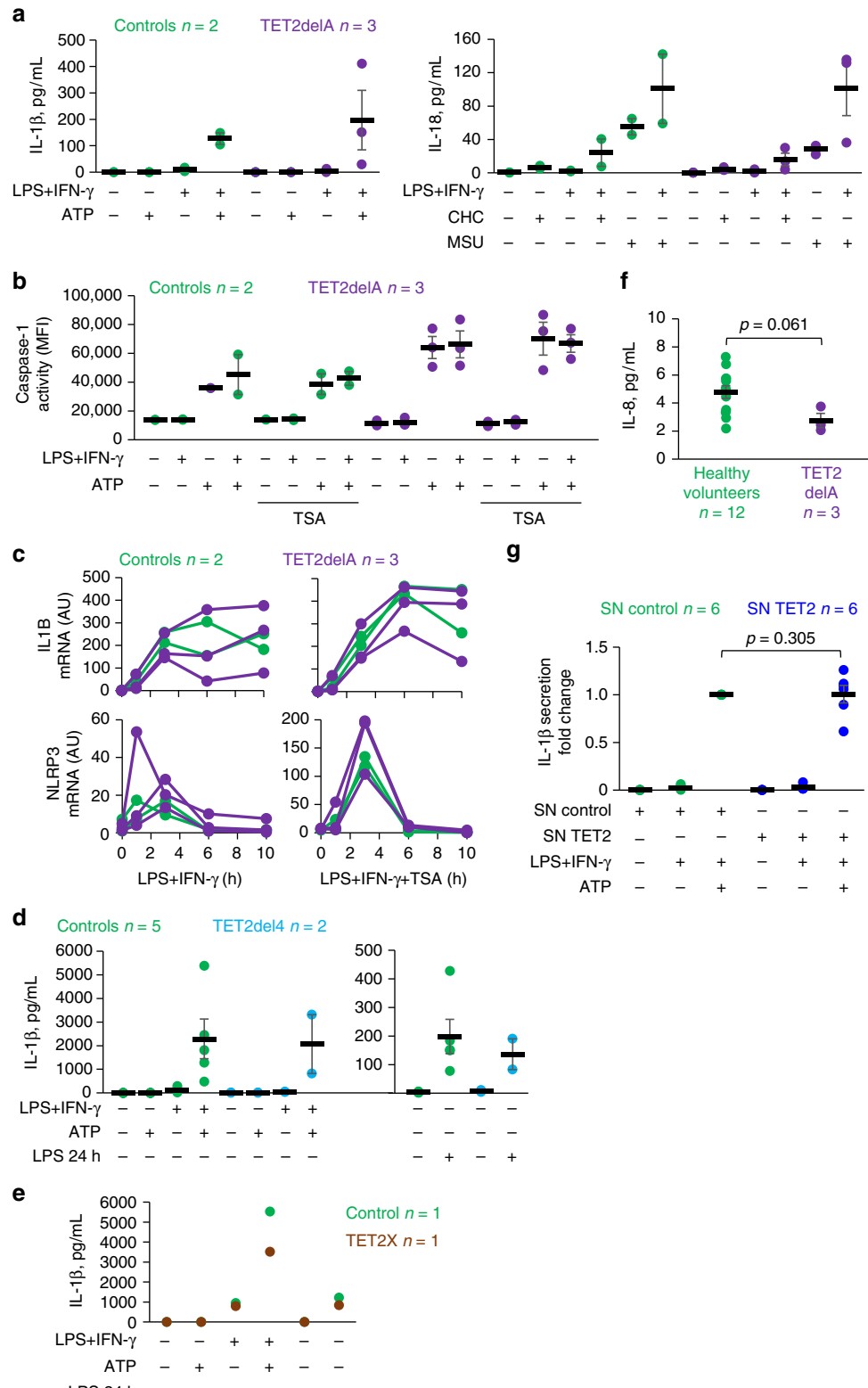

TET2 knockdown by siRNA in human blood monocyte-derived macrophages did not affect IL-6 expression (Supplementary Fig. 19f).

**Clinical evaluation of atherosclerosis**. Detailed clinical information was available from the eight Finnish *TET2* mutation carriers. The mother of Ly1, Ly2, and Ly3, who was a carrier of TET2delA, died at age 74 years due to pneumonia, with meningioma as a contributing condition according to her death certificate. Her mutation-free spouse had died of a presumed myocardial infarction at the age of 45. Ly1, Ly2, and Ly3 all had received radiotherapy as well as chemotherapy on one to three different occasions (Supplementary Table 1). Ly1 has a history of celiac disease and hypercholesterolemia (~8.5 mmol/l prior to medication) and coronary disease was diagnosed at age 62. Ly2

**Fig. 6** Inflammasome-mediated cytokine response and plasma IL-8 are not elevated in constitutional heterozygous *TET2* loss. **a–c** Monocyte-derived macrophages from TET2delA mutation carriers and controls were **a**, **b** primed for 6 h with lipopolysaccharides (LPS) and interferon-gamma (IFN-gamma) followed by NLRP3 inflammasome activation with ATP, cholesterol crystals (CHC), or monosodium urate crystals (MSU), or **c** treated with LPS and IFN-gamma ± trichostatin A (TSA) for the depicted times. **d**, **e** Monocytes from TET2del4 and TET2X mutation carriers and controls and **g** macrophages cultured from normal human donors and transfected with control or *TET2* siRNAs (SN) were stimulated with LPS + IFN-gamma + ATP as in **a** and **b**. **a**, **d**, **e**, **g** Secretion of mature interleukin-1 beta (IL-1B) and IL-18 was measured by enzyme-linked immunosorbent assay (ELISA) from cell culture supernatants, **b** cellular caspase-1 activity was detected by flow cytometry (MFI, median fluorescence intensity), and **c** mRNA expression was analyzed by quantitative PCR (AU, arbitrary units). **f** Plasma levels of the CXC chemokine IL-8 were measured by ELISA from TET2delA mutation carriers and healthy volunteers. Individual data points from each donor are shown with mean values ± s.e.m. **f**, **g** Statistical analysis was performed with Wilcoxon rank sum test

has been diagnosed with rheumatoid arthritis, hypercholesterolemia (total ~5 mmol/l, high-density lipoprotein ~ 0.7 mmol/l, low-density lipoprotein ~ 3.5 mmol/l prior to medication) and hypertension. He had smoked 10 pack years but at the age of 63 remains free of clinically manifest atherosclerotic disease. Ly3 died of septic *Neisseria meningitidis* meningitis free of CVD at the age of 50. Ly9 is a 41-year-old female ex-smoker, who carries TET2delA and has remained in good health. Ly11 is a 35-year-old female TET2delA carrier, diagnosed with mitochondrial encephalomyopathy, lactic acidosis, and stroke-like episodes syndrome, inherited from her mother. She has remained free of atherosclerotic disease. Ly14 is a 19-year-old healthy female smoker carrying TET2delA mutation. Id1 is a 21-year-old male who carries de novo TET2X mutation, free of CVD.

We then re-evaluated the existing radiographic data of the Finnish *TET2* mutation carriers. Computed tomography (CT) scans of chest and abdomen revealed calcified plaques in Ly1 and Ly2 at age 47 and 48, respectively, and thereafter, whereas Ly3 was lesion-free in five (abdominal, body, and three occasions chest) CTs between 46 and 48 years of age. Next, we performed carotid intimal ultrasound to all six living Finnish *TET2* mutation carriers. Measured mean and maximum far wall carotid intima-media thickness values from the right and left common carotid artery were evaluated in light of reference values derived from 740 Northern Finland Birth Cohort 1966 study subjects free from vascular disease or cardiovascular risk factors (K.K., personal communication). Non-obstructive atherosclerotic plaques as well as increased intima thickness in carotid arteries were detected in two subjects (Ly1 and Ly2, 65 and 63 years of age, respectively). In the four younger subjects (Ly9, Ly11, Ly14, and Id1), carotid ultrasound was normal.

## Discussion

In this study, we identified a family with heterozygous TET2 loss in seven individuals. Three of these individuals had been diagnosed with NLPHL. In addition, an individual with a de novo *TET2* mutation was identified from NFID Cohort data. In our genome-wide bisulfite sequencing analyses, significant global hypermethylation was seen in blood of *TET2* mutation carriers as compared to *TET2* wild-type cases. Hypermethylation was elevated especially at active enhancer regions. Our findings establish *TET2* as a gene predisposing to familial DNA demethylation deficiency and hematological aberrations when constitutionally mutated. We propose that the perturbation in hematopoiesis caused by reduced TET2 function appears to relate to aberrations in lineages that require synergistic actions of TET2 and master TFs involved in hematopoiesis.

Unlike many other lineage-specifying TFs, e.g CEBPα/β and PAX5, RUNX1/2/3 and PU.1 belong to TFs that have binding preference to unmethylated DNA[28]. Interestingly, TET2 has been shown to directly interact with RUNX1[29] and PU.1[30,31]. Our data provided evidence that TET2-mediated DNA demethylation is required at binding sequences of methyl-sensitive TFs involved in hematopoiesis, such as RUNX1/2/3 and PU.1, and that loss of one

allele has a significant effect on lineage-specific methylation levels. When cellular TET2 activity is reduced, the concerted action of methyl-sensitive TFs and TET2 may become compromised, leading to increased methylation in key regions for lineage development. It is important to note that even though we measured DNA methylation in circulating blood cells, the methylation differences that had emerged during differentiation remained in the progeny, and thus could be observed.

Based on regulon-level analysis of single-cell RNA-sequencing data, we found significantly lowered activity of the T-box TFs T-box 21 (TBX21) and eomesodermin (EOMES) in circulating CD8+ T cells and NK cells of the cancer-free TET2delA carriers. These TFs play crucial and partly overlapping roles in the development, activation, cytotoxic effector functions, and exhaustion of CD8+ T cells and NK cells, as well as in the development and maintenance of the CD8+ T memory cell pool[32–36]. These findings may imply dysregulation of cytotoxic lymphocyte maturation and function, and are compatible with the changes observed in detailed T-cell phenotyping by flow cytometry, where reduced CD8+ T effector memory cells and increased CD8+ exhausted TEMRA phenotype were detected in the TET2delA carriers. Interestingly, TBX21 has been shown to recruit TET2 to the lineage-specific signature gene *IFNG* during CD4+ T helper 1 cell differentiation, resulting in increased 5-hmC deposition at *IFNG* promoter and (−6) enhancer region[37]. We could show significantly increased methylation in TET2delA carriers at binding sequences of TBX21 in NK cells. Thus, reduced TBX21- and EOMES-mediated transcription in *TET2* mutation carriers could conceivably stem from inadequate recruitment of TET2 to gene regulatory regions by these TFs, although this remains to be studied in detail.

We observed decreased methylation relative to all open chromatin regions in the four *DNMT3A* germline mutation carriers in the binding regions of master TFs involved in hematopoiesis, suggesting that these regions are targets of the joint activity of TET2 and DNMT3A. When DNMT3A action is reduced, oxidation activity of TET2 leads to reduced levels of 5-mC, and emergence of 5-hmC and other TET2 oxidation products (toward unmethylated cytosine), as compared with normal state. Because this oxidation is a multistep cascade, the clearance of methylation in these TF-binding regions is likely strongest in hematological malignancies with concomitant mutations in *TET2* and *DNMT3A* that indeed are known synergistic drivers of neoplasia[38].

Recently, it was proposed that by studying DNA methylation in late-replicating domains, accumulation of cell divisions and cellular aging could be measured[16]. We did not observe a difference in methylation of these domains between *TET2* mutation carriers and their age-matched wild-type controls, suggesting similar history of cell divisions in both groups. Thus, both the clinical as well as molecular phenotype of the study subjects are compatible with low neoplastic potential of an individual $TET2^{+/−}$ cell. Absence of second hits imply little additional selective value for loss of both TET2 alleles and, consequently, rare emergence of *TET2*-null cells in humans. This is compatible with findings on

CH[1]. The emergence of *TET2*-mutant clones in the elderly may thus reflect increased resistance to senescence rather than strong expansive capability, as proposed also by others[39].

CVD is the most common cause of death in the Finnish population (Supplementary Data 7), radio- and chemotherapy promote atherosclerosis, and also other risk factors such as positive family history from the paternal side, hypercholesterolemia, and smoking were present in the TET2delA family. Considering these, CVD burden in the Finnish mutation carriers was unremarkable, and we were unable to derive significant clinical evidence for a role of *TET2* germline mutation positivity in predisposition to atherosclerosis. This result needs to be interpreted with appropriate caution as the number of individuals examined was small. CH is associated with CVD[3,4], smoking[39,40] as well as age, the latter two associating with inflammation and CVD. In two studies, association between CH and atherosclerosis-related conditions remained after adjusting for smoking[3,4]. In two other studies, Genovese et al.[40] found that excess of mortality in CH patients was explained by a combined effect of hematological malignancy and smoking, and Zink et al.[39] found no significant association between CH and smoking-associated disease after adjusting for smoking. A more recent study[41] found no evidence of association of *TET2*- and *DNMT3A*-driven CH to CVD in a female cohort. CH and CVD are both conditions that conceivably associate with quantity of smoking exposure, not accounted for in the publications thus far, and strongly with age. An individual's aging process is not directly reflected by years, but also nutrition, lifestyle, environment, and genetics play a role[42]. It is difficult to perfectly adjust for these factors, and thus separating two age-related traits in association analyses is challenging.

Mechanistically, exacerbated IL-1β secretion by macrophages was proposed to be essential for accelerated atherosclerosis in context of CH due to *Tet2* loss in mice[5]. However, a more recent study could not show increased IL-1β secretion in *Tet2*$^{-/-}$ mouse macrophages[4]. Instead, increased CXC chemokine secretion was proposed as the key factor for accelerated atherosclerosis[4], with increased IL-8/CXCL8 secretion suggested as the possible effector in humans. We did not find evidence supporting either mechanism in the six *TET2* germline mutation carriers examined, representing three different truncating mutations. Our results as such are not discrepant with the mouse work, as the clinical and molecular consequences of TET2 loss in mice do not need to be identical to the human phenotype. The effects of TET2 loss are mediated through hypermethylation within a genome, and the two genomes were separated tens of millions of years ago. For example, mice are devoid of TEMRA cells[43], which we found to be elevated in the elderly *TET2* mutation carriers and may influence the human phenotype.

The single-cell RNA-sequencing data revealed significantly elevated expression of *CXCR4* in all blood cells of the TET2delA individuals, especially in T and NK cells. This chemokine receptor mediates leukocyte trafficking in response to its ligand CXCL12 and has crucial roles e.g. in B-cell development[44]. Notably, CXCR4 knockdown/knockout in bone marrow or in vascular wall cells aggravates the progression of atherosclerosis in mouse models[45,46]. Moreover, a common human *CXCR4* variant, C-allele at rs2322864, is significantly associated with increased risk of coronary heart disease and with reduced *CXCR4* expression in whole blood and in carotid atherosclerotic plaques[45]. Thus, the elevated *CXCR4* expression in blood cells of TETdelA carriers exemplifies one possible atheroprotective effect by TET2 loss that may help to counterbalance possible proatherogenic effects in humans. We could also detect reduced expression of immunoglobulin genes such as *IgJ* in B cells of cancer-free TET2delA carriers. It has been shown that *IgJ*, *Igκ*, and VDJ rearranged *IgH*

co-localize to the same transcription factories[47]. Furthermore, the expression of *IgJ* and *Igκ* is PU.1 regulated and *Igκ* is silenced in mouse early B cells with Tet2/Tet3 loss[31,48,49]. Altogether, these data suggest that *TET2* has a role in regulating immunoglobulin genes in human B cells, possibly through cooperation of TET2 and lineage-specific TFs, and/or through high *CXCR4* expression[50].

It is important to note that the exposure of the germline mutation carriers to effects of TET2 loss is life-long and extreme, as compared to the typical CH setting of a minor somatic heterozygous *TET2*-mutant subclone—2% or higher variant allele frequency[39]—emerging late in life. Our data raise the possibility, that circulating *TET2*-mutant cells may be of limited importance as an atherosclerosis risk factor, at least in the context of germline mutation where all cell types carry the mutation. While this finding does not support a major role of TET2 loss in atherosclerosis, it is equally important to note that molecular characters of CH cells that have accumulated additional changes during somatic selection may be more aggressive and atherogenic. If so, the detailed mechanisms need to be determined.

Vitamin C boosts TET2 function[51]. Thus, adequate vitamin C intake in the context of hereditary heterozygous *TET2* loss-of-function mutations might reduce risk of neoplasia, a subject that needs further study. At present, it appears appropriate to at least advise such individuals to ensure that the daily dose recommended for the general population is reached. In the context of *TET2*-mutation-positive CH, prophylactic treatment for atherosclerosis using cholesterol-lowering medications or compounds targeting IL-1β and other inflammatory pathways as proposed[2,4,5] would, if effective, be a significant advance. If ineffective, such an intervention would cause cost and side effects. Thus, it is important to more thoroughly scrutinize the proposed association between CH and CVD, and the possible mechanisms underlying it.

## Methods

**Study approval**. The samples and patient information were obtained with approval from the ethics committee of the Hospital District of Helsinki and Uusimaa, the Northern Ostrobothnia Hospital District, the Ethik Kommission Beider Basel as well as Finnish National Supervisory Authority for Welfare and Health. Informed consent was obtained for all participants. This study was conducted in accordance with all relevant rules and regulations, including the Declaration of Helsinki.

**Immunohistochemistry**. Histopathological analysis of NLPHL samples was performed by an experienced hematopathologist. Formalin-fixed paraffin-embedded (FFPE) samples were sectioned to 5 μm thickness, and stained with hematoxylin-eosin, and with antibodies against CD3, CD15, CD20, and CD30 according to standard procedures at Helsinki University Hospital Pathology Lab.

**Detection of genomic regions segregating with lymphoma**. Genomic DNA was extracted from blood samples of three family members, Ly1, Ly2, and Ly8. Blood sample from Ly3 could not be obtained, since the person is deceased, so a FFPE tissue was used for DNA extraction. Whole-genome SNP genotypes were obtained using HumanOmniExpress-FFPE BeadChip (Illumina, Inc., CA, USA) of 693,543 markers. All procedures were performed according to the manufacturer's instructions at Finnish Institute for Molecular Medicine (FIMM) Genome and Technology Centre. Genotype calling and quality control of the data were done with GenomeStudio Genotyping module version 1.9.4 (Illumina). A parametric linkage analysis with dominant inheritance model (50% penetrance) was performed using MERLIN[52] similar to Saarinen et al.[53]. Ly8 was considered as unaffected. Further germline variant analyses were focused on linked regions i.e. the segregating genomic regions with a positive logarithm of odds score (>0) and a length of at least 5 cM.

**Germline variant analysis**. Ly1 genomic DNA library was prepared and sequenced according to Complete Genomics (Complete Genomics Inc., Mountain View, CA, USA) paired-end sequencing service protocols. Complete Genomics' service was conducted with standard coverage: 40× average coverage and 90% callable diploid loci on the human reference genome. Both read alignment and variant calling were included in the service.

Whole-exome sequencing was performed for patients Ly2 and Ly3 by capturing exomic regions using SureSelect Human All Exon Kit v.1.0 (Agilent Technologies, Inc., CA, USA) and SeqCap EZ Human Exome Library (Roche Nimblegen, Inc., WI, USA), respectively, using the standard protocols. Paired-end sequencing was performed using Illumina Genome Analyzer II at the Finnish Institute of Molecular Medicine (Ly2) and Illumina HiSeq2000 at Karolinska Institutet (Sweden; Ly3). The read lengths were 82 bp (Ly2) and 76 bp (Ly3). The paired-end reads were quality-controlled using FastQC (www.bioinformatics.babraham.ac.uk/projects/fastqc), and then aligned to human reference genome version GRCh37 from the 100 Genomes Project (human_g1k_v37.fasta) using an in-house pipeline consisting of publicly available tools [bwa[54] (version 0.5.9) for alignment, Genome Analysis Toolkit (GATK)[55] (version 1.0.5083 for Ly2 and version 1.4 for Ly3) for realignment around indels, Picard tools (http://picard.sourceforge.net) for duplicate removal, and samtools[56] for sorting and indexing]. Default parameters were used except for bwa aln -q was set to 5 and -n to 0.06, and bwa sampe max insert size was set to 800.

We first determined the variants of patient Ly1 inside the linked regions using the whole-genome sample and a control set consisting of whole genomes and exomes of in-house Finnish patients' germline DNA. Additionally, publicly available data were used (UK10K, www.uk10k.org, and Sequencing Initiative Suomi, www.sisuproject.fi). In total, the control set consisted of 5197 whole genomes or exomes. Only variants with zero hits in the control set and Complete Genomics quality score (VarScoreVAF) higher than 60 were accepted into downstream analyses. Then, for each remaining variant position in the whole-genome set, we calculated the coverages of reference and variant reads from the whole-exome-sequenced patients Ly2 and Ly3, and classified each position based on coverage and visual inspection into three categories: (i) variant is shared by all three patients; (ii) variant is not shared by all three; and (iii) not enough evidence to say whether the variant is shared or not. If the variant read coverage in samples Ly2 and Ly3 was below four, we put the variant into category (iii). Variants belonging to sets (i) and (iii) were accepted into downstream analyses. The visual inspection of the alignments and control filtering was done using the visualization software BasePlayer[57].

Next, a functional effect prediction was done by two different algorithms (Provean[58] and SIFT[59]). Single-nucleotide or multiple single-nucleotide variants that were predicted damaging or deleterious by both algorithms, and indels and stop codon producing variants constitute the set of preliminary shared candidate variants (Supplementary Table 2).

Finally, we filtered the shared variant list further by using the Exome Aggregation Consortium (version 0.2) dataset consisting of 60,706 unrelated individuals' exomes by allowing no hits in the control set. We further limited the false positive calls down by using the strict genomic regions accessible to accurate analysis from the 1000 Genomes Project (phase 1). The remaining variants on the list were validated by Sanger sequencing. The deletion c.455_462delACAGCCAC in ABRA was found to be homozygous in the healthy mother of the patients Ly1, Ly2, and Ly3, excluding it from the final candidate variants (Supplementary Table 2). Furthermore, the TET2 c.4500delA mutation status of the mother was examined from archival tissue derived DNA, and she was found to carry the mutation. All primer sequences used in the validation of the candidate variants are available in Supplementary Table 13.

**Bisulfite sequencing library preparation and data processing.** The SureSelect target enrichment system covering 84.5 Mb (Agilent Technologies, Inc., CA, USA) was used to prepare bisulfite sequencing libraries from blood DNA of patients (Ly1 and Ly2), healthy family members (Ly8, Ly9, Ly10, Ly11, Ly12, Ly13, and Ly14), baseline controls (controls 1–5), and DNMT3A mutation carriers and their age-matched controls (Supplementary Table 3). In addition, blood DNA of a patient (HLRCC_N7) with germline fumarate hydratase (FH) mutation (c.1027C>T, p. R343X) was included in the study, because FH-null paragangliomas have been shown to display genomic hypermethylation[60]. Two separate bisulfite sequencing libraries of the patients Ly1 and Ly2 were prepared from DNA samples extracted in years 2007 and 2009. Sample preparations were done according to the manufacturer's instructions. Illumina paired-end sequencing for targeted libraries from Ly1, Ly2, Ly8, Ly9, Ly10, and Ly11 was done at Karolinska Institutet using 100 bp read length and the HiSeq2000 platform. Illumina paired-end sequencing for targeted libraries from Ly12, Ly13, Ly14, and DNMT3A mutation carriers and their age-matched controls from NFID cohort was done as a service at BGI (BGI Tech Solutions Co., Ltd., China) using 126 bp read length and the HiSeq2500 platform.

WGBS library preparations and Illumina sequencing were done as a service at BGI (BGI Tech Solutions Co., Ltd., China). Bisulfite treatment was done with EZ DNA Methylation-Gold Kit (Zymo Research, CA, USA) for 300–400 bp size-range fragments with methylated adapters in 5′ and 3′ ends. Sequencing was done with the HiSeq X-Ten platform using paired-end 150 bp read length.

Raw sequencing reads were quality and adapter trimmed with cutadapt in Trim Galore. Trimming of low-quality ends was done using Phred score cutoff 30. Adapter trimming was performed using the first 13 bp of the standard Illumina paired-end adapters. Read alignment was done against hg19/GRCh37 reference genome downloaded from UCSC Genome Browser with Bismark (version v0.14.4)[61] and Bowtie 2 (version 2.2.5)[62]. Duplicates were removed using the Bismark deduplicate function. Extraction of methylation calls was done with Bismark methylation

extractor (v0.14.4) discarding first 10 bp of both reads and reading methylation calls of overlapping parts of the paired reads from the first read (--no_overlap parameter).

**Analysis of targeted bisulfite sequencing data.** Detection of DMCs and DMRs was done with the DSS[63] R package (version 2.14.0). Only cytosines with minimal depth of coverage of six were included. Methylation values were smoothed in 300 bp windows before DMC detection. DMCs were detected on autosomes with Wald test requiring methylation difference >0.2 and posterior probability >0.99999 in comparison of each sample against the five baseline control samples (controls 1–5). For the comparison of number of hyper- or hypomethylated CpGs sites between five TET2 mutation carriers and five non-carriers, two-tailed Wilcoxon rank sum tests were performed. DMRs were detected by comparing (i) TET2delA carriers (Ly1, Ly2, Ly9, Ly11, and Ly14) to wild-type and control samples (Ly8, Ly10, Ly12, Ly13, HLRCC_N7, and controls 1–5) as well as by comparing (ii) NLPHL cases (Ly1 and Ly2) to cancer-free TET2delA carriers (Ly9, Ly11, and Ly14). DMRs were called requiring minimum number of three CpG sites, minimum length 50 bp, and p-value < 0.01 for (i) and <0.05 for (ii).

ATAC-seq open chromatin regions and fragment counts were downloaded from GEO accession GSE74912 for hematopoietic cell types[17]. One-sided Wilcoxon rank sum test and FDR-adjusted p-value ≤ 0.05 were used to identify cell-type-specific regions of open chromatin similar to Farlik et al.[64]. Genomic sites recognized by each TF (Supplementary Fig. 12)[28,65,66] were searched from the human genome hg19 using program MOODS[67] with p-value cutoff of $10^{-4}$ and score cutoff of 5. Open chromatin regions and motif match sites were compared with BEDtools (v2.26.0) intersect requiring full overlap. The statistical significance of methylation increase at open chromatin regions with a motif match was assessed with a permutation test for each cell type. The test statistic was the average methylation difference at the open chromatin regions between 5 TET2 mutation carriers, and 10 merged non-carriers and controls. Motif matches were permuted by randomly repositioning each motif match within the cell-type-specific open chromatin regions using BEDtools (v2.26.0) shuffle. A total of 1000 permutations were performed for each motif and cell type, and an empirical p-value calculated as the proportion of the cumulative distribution where the random test statistic was higher than the observed methylation difference. For confirmation, 10,000 permutations were also performed for the SPI1 (PU.1) and RUNX2 binding sequences at each of the cell-type-specific open chromatin regions.

The average methylation values at cell-type-specific open chromatin regions displayed in Fig. 4a were calculated by taking into account CpG sites with minimum depth of coverage of 6. Two-sided Wilcoxon rank sum test was used to test the difference of these average cell-type-specific methylation values between TET2 mutations carriers (Ly1, Ly2, Ly9, Ly11, and Ly14), and 10 wild-type and control samples (Ly8, Ly10, Ly12, Ly13, HLRCC_N7, and controls 1–5).

Pearson's correlation displayed in Fig. 4e was calculated for methylation changes in mutation carriers as compared to controls at open chromatin regions with master TF-binding sequence, relative to all open chromatin regions in each of the 10 cell types. Methylation values from TET2delA carriers (Ly1, Ly2, Ly9, Ly11, and Ly14) were compared to 10 wild-type and control samples (Ly8, Ly10, Ly12, Ly13, HLRCC_N7, and controls 1–5), whereas methylation values from DNMT3A mutation carriers (Id5, Id7, Id9, and Id11) were compared to four controls from NFID cohort (Id6, Id8, Id10, and Id12). Pearson's product-moment correlation test was performed in R resulting in t = 2.2377, df = 48, and p-value = 0.02992.

**Analysis of WGBS data.** Detection of DMCs was done with the DSS[63] R package (version 2.14.0) at cytosines with minimal depth of coverage of 2. Methylation values were smoothed in 300 bp windows before DMC detection. DMCs were detected on autosomes with Wald test requiring methylation difference >0.1 and posterior probability >0.99.

Enrichment of hyper- and hypomethylated DMCs were analyzed with Locus Overlap Analysis (LOLA)[68] R package (version 1.8.0) using Roadmap Epigenomics data (https://egg2.wustl.edu/roadmap/data/byFileType/peaks/consolidated/narrowPeak/) and UCSC features provided in the LOLA extended and core databases (version 170206), and using differentially methylated and hydroxymethylated regions from Zhang et al.[19] that were converted from mm9 to hg19 coordinates with liftOver. All CpG sites that were included in the DMC testing were used as the background set.

Coordinates of Solo-WCGW CpGs [CpGs with the combination of zero neighboring CpGs ("solo") and flanked by an A or T ("W") on both sides] in common PMDs were downloaded from Zhou et al.[16]. Boxplots were generated by considering methylation percentage at CpG sites with minimum depth of coverage of 6.

**Immunophenotyping of mutation carriers.** Lymphocyte, B-, and T-cell phenotyping were performed as previously published[69–71] as a service at Helsinki University Hospital (analysis codes: 6258smB-Fc, 21388Tdif-Fc, and 8302LyDiff-T). Briefly, for flow cytometry, cells in EDTA samples were stained using whole blood technique, and analyzed on a FACSCanto II flow cytometer (Beckton Dickinson Biosciences, San Jose, CA, USA). Fresh heparin-blood samples or peripheral blood mononuclear cells (PBMCs) were used for B- and

T-lymphocyte immunophenotyping. Four- or 10-color flow cytometry panel with monoclonal antibodies (mAbs) against the surface antigens IgM, IgD, CD3, CD4, CD8, CD16/56, CD19, CD21, CD27, CD33, CD34, CD38, CD45, CD56, CD57, CD133, HLA-DR, CD62L, CD45RA, and CD45RO (BD Biosciences, San Jose, CA)[70]. The memory status of T cells was studied with the antibody panel including anti-CD45, -CD3, -CD4, -CD8 -CD45RA, and -CCR7 (R&D Systems, Minneapolis, MN, USA)[70,71].

**Single-cell transcriptome sequencing and analysis**. Whole blood was collected in EDTA-treated collection tubes. Red blood cells were lysed by adding a volume of ACK lysing buffer (Gibco, New York, NY, US) corresponding to 10–20 times the volume of the blood sample and incubating at room temperature (RT) for 3 min. White blood cells were collected by centrifugation at $300 \times g$ for 5 min at RT. The cells were resuspended in cold phosphate buffered saline (PBS) corresponding to five times the volume of the whole-blood sample and collected by centrifugation at 4 °C. After resuspending in cold 0.04 % bovine serum albumin/PBS, the cells were filtered with FlowMiTM Tip strainer of 40 μm porosity (SP Scienceware, Bel-Art Products, Wayne, NJ, US). Finally, the cells were stained with Trypan Blue (Invitrogen, Waltham, MA, USA), counted with Countess Automated Cell Counter (ThermoFisher Scientific), and adjusted to the final concentration of $1 \times 10^6$ cells/ml.

Single cells were captured into 10x barcoded gel beads and RNA-sequencing library preparation was done using Chromium Single Cell 3′ v2 chemistry (10x Genomics, Pleasanton, CA, USA) at FIMM Single Cell Analytics core facility (Finland). Sequencing was performed as recommended with 98 bp length of read 2 using HiSeq4000 sequencer (Karolinska Institutet, Sweden). Capture and sequencing statistics are presented in Supplementary Table 14. Cell Ranger pipeline (v.2.0.0, 10x Genomics) was used for data processing with default parameters using prebuild human hg19 genome reference provided by 10x Genomics. Cell types were identified from $K$-means ($K = 10$) clusters utilizing CD3D, NKG7, IL7R, CD79A, CST3, S100A9, and LYZ marker genes (Supplementary Fig. 14). Data were separately combined from all eight samples and from six cancer-free family members with cellranger aggr pipeline. Statistical tests between cancer-free TET2 mutation carriers and wild-type individuals were performed in Loupe Cell Browser (v1.0.3, 10x Genomics) implementing a variant of the negative binomial exact test, and for large counts, the fast asymptotic beta test. Resulting p-values were adjusted for multiple testing using the Benjamini-Hochberg procedure. Twenty most up- and downregulated genes between cancer-free TET2delA carriers and wild-type individuals are presented for each major cell type based on fold change regardless of statistical significance (Supplementary Tables 9–12). XIST and RPS4Y1 were among the most significantly differentially expressed genes due to gender bias between TET2delA carriers (100% female) and wild-type individuals.

SCENIC[72] was used to identify changes in activity of TFs and their target genes between cancer-free TET2delA carriers (Ly9, Ly11, and Ly14) and their age-matched wild-type family members (Ly8, Ly10, and Ly13). In brief, SCENIC calculates sets of genes that are co-expressed with known TFs and further retains target genes that show significant motif enrichment of the correct upstream regulator in cis in the final co-expression modules, "regulons". The analysis was limited to TFs that had expression in at least 1% of the cells and at least 199.92 reads (the total number the gene would have, if it was expressed with a value of 3 in 1% of the cells) across all samples, and whose binding motifs were included in the species-specific databases for RcisTarget (motif collection version 9 including 24,000 motifs). In particular, we used the databases that score the motifs in gene promoters (up to 500 bp upstream the TSS), and in the 20 kb flanking the TSS (±10 kbp). First, we calculated SCENIC regulons using single-cell transcriptome data from the wild-type family members only. Next, activity of each regulon in each cell was estimated with the AUCell method implemented in SCENIC for both TET2delA and wild-type samples. Comparisons of the regulon activities between cells from TET2delA carriers and age-matched family members were done with Wilcoxon rank sum test. Statistical significance for differential activity was calculated for 105 TFs that had enough expression and binding motifs available.

**CH detection**. Deep exome sequencing was performed from blood DNA samples of five living TET2 mutation carriers and three wild-type family members extracted at multiple time points (Ly1 and Ly2 samples from years 2007, 2009, and 2017; Ly8, Ly9, and Ly11 samples from years 2007 and 2017; Ly10 sample from year 2007; and Ly13 and Ly14 samples from year 2017). Library preparations were performed with SeqCap EZ Exome v3 (Roche, Switzerland) using six different index primers per sample for which paired-end Illumina sequencing was done with 75 bp read length and HiSeq4000 sequencer at Karolinska Institutet (Sweden). After alignment (bwa version 0.7.12), base recalibration (GATK 3.5), realignment around indels (GATK 3.5), and duplicate removal (MarkDuplicates; Picard Tools version 1.79), data from libraries with six different indexes were merged together for variant calling. The merged libraries had >200× average coverage at the capture target regions (Supplementary Table 15). Due to low allele frequency of variants expected as indicative of CH[40], Unified Genotyper from GATK (version 3.5) was used to genotype all variants reported in the Catalogue Of Somatic Mutations In Cancer (COSMIC) v80 (release 20170213). Variants were searched from the strict

genomic regions accessible to accurate analysis (the 1000 Genomes Project Phase 1) excluding variants with StrandOddsRatio > 3.

**Functional analyses of human monocytes and macrophages**. Functional analyses were performed (1) in macrophages cultured from fresh blood samples of four lymphoma-free members of the TET2delA family (Ly8+/+, Ly9+/−, Ly11+/−, and Ly14+/−) and from an unrelated female control, (2) in monocytes isolated from frozen PBMCs of two TET2del4 mutation carriers (p226 and p302) and five unrelated controls, and (3) in monocytes isolated from fresh blood samples of the carrier of the de novo TET2X mutation (Id1) and a mutation-negative first-degree relative as control (Id4). TET2 knockdown experiments were performed in macrophages cultured from buffy coat preparations from anonymous blood donors (Finnish Red Cross Blood Service, Finland).

PBMCs were isolated by density gradient centrifugation in Ficoll-Paque PLUS (GE Healthcare, Chicago, IL, USA). For macrophage culture, PBMCs were adhered to culture plates for 1 h, washed three times in PBS, and the adherent monocytes were differentiated into macrophages via 7 days culture in Macrophage-SFM medium (Gibco) supplemented with penicillin-streptomycin (Gibco) and 10 ng/ml recombinant human granulocyte–marophage colony-stimulating factor (Miltenyi Biotec, Germany). Monocytes were isolated from PBMCs by negative selection using the Human Pan Monocyte Isolation Kit (Miltenyi Biotec) and magnetic-activated cell sorting in QuadroMACS (Miltenyi Biotec) separator according to the manufacturers' protocols; monocytes were allowed to rest overnight in Macrophage-SFM medium (Gibco) supplemented with penicillin-streptomycin (Gibco) before performing the stimulations. For inflammasome activation, cells were first primed for 6 h with 1 μg/ml LPS from Escherichia coli O111:B4 (Sigma-Aldrich, St. Louis, MO, USA) and 5 ng/ml (=100 IU/ml) recombinant human interferon-γ (ImmunoTools GmbH, Germany), in the presence or absence of 1 μM TSA (InvivoGen, San Diego, CA, USA) where indicated. After priming, inflammasome activation was triggered by stimulation with 5 mM ATP (Sigma-Aldrich) for 45 min, 1 mg/ml cholesterol crystals for 16 h, 0.2 mg/ml monosodium urate (MSU) crystals for 16 h, or by intracellular transfection of 2 μg/ml UltraPure LPS from E. coli O111:B4 (InvivoGen) with 5 μl/ml Lipofectamine2000 (Invitrogen) for 5 h. Cholesterol (Sigma-Aldrich) at 12.5 g/l in 95% ethanol was heated to 60 °C, sterile-filtered, crystallized at RT, dried, and ground to a size range of 1–10 μm using a sterile mortar and a pestle. Uric acid (Sigma-Aldrich) at 5 g/l in deionized water was heated to 60 °C, adjusted to pH 8.9 with 0.5 N NaOH, and crystallized at RT. The MSU crystals were recovered by centrifugation, washed, dried, and sterilized at 180 °C. Endotoxin levels of the cholesterol and MSU crystals were below the detection limit (<0.03 EU/ml) of Pyrogent gel clot LAL assay (Lonza, Switzerland). Monocyte inflammasome response to prolonged LPS exposure was studied by 24 h stimulation with 1 μg/ml LPS from E. coli O111:B4 (Sigma-Aldrich, St. Louis, MO, USA). For TET2 knockdown, macrophages were transfected twice (on differentiation days 4 and 5) with 100 nM pool of four TET2-targeted siRNAs (Human TET2 GeneSolution siRNA, Qiagen; Supplementary Table 16) using the HiPerFect transfection reagent (Qiagen) at 10 μl/ml.

Mature, cleaved forms of IL-1β/IL-18 were detected from cell culture media supernatants using Human IL-1β/IL-1F2 DuoSet ELISA and Human Total IL-18 DuoSet ELISA, and IL-8 from plasma using Human IL-8/CXCL8 Quantikine HS ELISA Kit (all from R&D Systems). For quantitative real-time PCR, RNA was isolated using RNeasy Plus Mini Kit (Qiagen, Germany), followed by cDNA synthesis with iScript kit (Bio-Rad Laboratories, Inc., Hercules, CA, USA). Quantitative PCR was performed from 10 ng of cDNA per reaction using LightCycler480 SYBR Green I master (Roche) and LightCycler96 instrument (Roche). See Supplementary Table 16 for the primer sequences. The data were analyzed using LightCycler96 SW 1.1 software (Roche) and relative gene expression was calculated using the $2^{(-\Delta\Delta Ct)}$ method with RPLP0 used as the endogenous reference gene. For detection of active caspase-1, primed macrophages were detached from culture dishes using Accutase Cell Detachment Solution (EMD Millipore, Billerica, MA, USA), resuspended in culture medium and stimulated with 5 mM ATP for 15 min at +37 °C. FAM-FLICA Caspase-1 Detection Kit (ImmunoChemistry Technologies, LLC., Bloomington, MN, USA) was then used according to the manufacturer protocol. Briefly, 1× FAM-FLICA caspase-1 substrate was added for 30 min at +37 °C, followed by two washes in 1× Apoptosis Wash Buffer and staining with propidium iodide. The cells were analyzed immediately using BD Accuri C6 flow cytometer (BD Biosciences; instrument maintained by the Biomedicum Flow Cytometry Unit) and the data are expressed as median fluorescence intensity.

RNA-sequencing libraries from monocyte-derived macrophages from three individuals with heterozygous TET2 loss (Ly9, Ly11, and Ly14) and two wild-type controls (Ly8 and an unrelated control) were prepared using ScriptSeq RNA-Seq Library Preparation Kit and Illumina sequenced with paired-end 75 bp reads at FIMM (Finland). Reads were quality- and adapter-trimmed with cutadapt version 1.3 in Trim Galore using default parameters. The number of paired-end reads after trimming varied from 44 to 57M. RNA-seq data were preprocessed using Kallisto (version 0.43.1) software[73]. Kallisto quantifications were ran in paired-end and fr-stranded mode with 100 bootstraps against the GENCODE transcript sequences (Release 26 mapped to GRCh37). The quantification results were normalized based on the sleuth (version 0.28.1) R package[74]. Differentially expressed genes between TET2 mutants and controls in different conditions were calculated with Wald test

implemented in sleuth. PANTHER (version 12.0 Released 2017-07-10) protein class statistical overrepresentation test (release 20170413) was applied to genes with $q$-value < 0.1, and $b$-value (the "beta" value, analogous to fold change) >0 or <0. To check *TET2* c.4500delA allele frequencies, trimmed reads were also aligned with HISAT2[75] (version 2.1.0) to GRCh37 Ensembl gene annotations.

**TET2 western blot**. Proteins were extracted from Epstein-Barr virus (EBV)-transformed lymphoblastoid cells with RIPA buffer (Sigma-Aldrich) supplemented with proteinase inhibitor (Roche). Twenty micrograms of protein were loaded into a 10% Tris-HCl gel (Bio-Rad). A mouse mAb against N-terminal TET2 (1:500 dilution, hT2H21F11, #MABE462; Merck, Kenilworth, NJ, USA) and rabbit polyclonal antibody against vinculin (1:1000 dilution, H-300, sc-5573; Santa Cruz Biotechnology, Dallas, TX, USA) was used, at 4 °C overnight incubation. Secondary antibodies against mouse and rabbit were incubated at RT for 2 h. Proteins were visualized by Amersham™ ECL Prime Western Blotting Detection System (GE Healthcare). Immunoblots were quantified with ImageJ software (http://imagej.nih.gov/ij/). Full gel image of an extended TET2 western blot run with two different amount of protein extract is shown in Supplementary Fig. 3a.

**Chromatin immunoprecipitation**. Briefly, EBV-transformed lymphoblastoid cells from three carriers of TET2delA (Ly9, Ly11, and Ly14) and two wild-type family members (Ly8 and Ly10) were fixed in 1% formaldehyde for 10 min at RT and reaction was stopped by adding 0.125 M glycine. Cells were washed in ice-cold PBS twice and then resuspended in lysis buffer (5 mM PIPES (pH 8.0), 85 mM KCl, and 0.5% NP-40). Lysate was then resuspended in 800 µl of RIPA buffer (1% NP-40, 0.5% sodium deoxycholate, and 0.1% SDS in PBS) containing protease inhibitors (Roche). Chromatin was sonicated to an average fragment length of 150–500 bp using Sonicator Misonix S-4000 with the following parameters: amplitude 30%, pulse-on time 30 min, pulse-off time 60 min, for a total sonication time of 10 min. Samples were centrifuged at $13,200 \times g$ for 5 min at 4 °C twice to collect the supernatant. Dynabeads protein-A magnetic beads (Invitrogen) were washed with 0.05% Tween-20 in PBS. Eight micrograms of antibody Anti-Histone H3 (acetyl K27) (ab4729; Abcam, UK) were added to the beads and incubated for 15 min on a rotator at RT. For each sample, 60 µl of the sonicated chromatin were stored as input fraction, and 750 µl were incubated with the antibody-coupled magnetic beads on a rotator overnight at 4 °C. After incubation, beads were washed twice with low-salt washing buffer (0.1% SDS, 1% Triton X-100, 2 mM EDTA, 20 mM Tris-HCl (pH 8.0), and 150 mM NaCl), twice with high-salt washing buffer (0.1% SDS, 1% Triton X-100, 2 mM EDTA, 20 mM Tris-HCl (pH 8.0), and 500 mM NaCl), once with lithium chloride washing buffer (1% NP-40, 1% sodium deoxycholate, 1 mM EDTA, 20 mM Tris-HCl (pH 8.0), and 500 mM LiCl), and twice with 1× TE (10 mM Tris-HCl (pH 8.0) and 1 mM EDTA). Antibody-bound chromatin samples were eluted from the beads by incubating for 1 h at 65 °C in 200 µl of IP elution buffer (1% SDS, 0.1 M NaHCO3, and 10 mM Tris-HCl (pH 7.5)) and crosslinking was reversed by overnight incubation at 65 °C. The eluted DNA was purified using phenol-chloroform method, followed by library preparation for Illumina HiSeq Rapid paired-end 60 bp sequencing at FIMM (Helsinki).

Raw sequencing reads were quality and adapter trimmed with cutadapt version 1.16 in Trim Galore version 0.3.7 using default parameters. Trimmed reads were aligned to hs37d5 reference genome using Bowtie 2[62] (version 2.1.0). Duplicate reads were removed with samtools rmdup (v1.7) after which 33–41M nonredundant reads were left in ChIP and input samples (Supplementary Table 17). Fragment coverage of paired-end reads was calculated from bam files with BEDtools genomecov (v2.26.0). Difference of ChIP fragment enrichment (ChIP fragment coverage − input fragment coverage) as compared to WGBS methylation difference between TET2delA carriers and wild-type individuals was calculated at CpG islands (cpgIslandExt) downloaded from UCSC. Only CpG islands with average ChIP fragment enrichment in wild-type samples ≥10 were included in correlation testing and in Fig. 3c. Pearson's product-moment correlation test was performed in R resulting in $t = -4.7389$, df = 6770, and $p$-value = 2.192e-06.

**Material availability**. Samples from the NFID cohort are stored in a biobank and available through an access committee. The other samples have been collected for the research groups (L.A.A. and R.C.S.). Thus, for the latter any request would need to be evaluated individually in light of the appropriate local rules and regulations and unrestricted access cannot be guaranteed.

## Code availability

Code for performing the permutation analysis is available upon request.

## Data availability

All genetic, expression, methylation, and ChIP-seq data used in the study and in Figs. 2–5 has been submitted to European Genome-phenome Archive (EGA), which is hosted at the EBI and the CRG, under accession number EGAS00001003454 for long-term archival. Data that potentially allow identification of individuals must be protected, and thus, an access committee has been established. ATAC-seq open chromatin regions and

fragment counts for hematopoietic cell types were downloaded from GEO accession GSE74912. Roadmap Epigenomics data provided in the LOLA extended databases (version 170206) were downloaded from [http://cloud.databio.org/regiondb/]. The human-specific databases for RcisTarget were downloaded from [https://resources.aertslab.org/cistarget/databases/homo_sapiens/hg19/refseq_r45/mc9nr/gene_based/hg19-500bp-upstream-7species.mc9nr.feather] and [https://resources.aertslab.org/cistarget/databases/homo_sapiens/hg19/refseq_r45/mc9nr/gene_based/hg19-tss-centered-10kb-7species.mc9nr.feather] with R version 3.5.0.

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

## Acknowledgements

We thank Alison Ollikainen, Inga-Lill Svedberg, Sini Marttinen, Iina Vuoristo, Jiri Hamberg, Heikki Metsola, and Sirpa Miettinen for excellent technical assistance, and Professor Riitta Herva from the Pathology Department of University of Oulu for providing us with NLPHL samples. We are also greatly indebted to the study subjects for participating in this study. This work was supported by grants from the Academy of Finland [Finnish Centre of Excellence Programs (#250345 and #312041), and personal grants for O. Kilpivaara (#137680 and #274474) and for P.V. (#260370)], the Finnish Cancer Society, the Sigrid Juselius Foundation, Maire Lisko Foundation (personal grant for K.R.), Reumatautien tutkimussäätiö (personal grant for K.R.), Finnish Foundation for Pediatric Research & Pediatric Research Center, Helsinki University Hospital Research Funds, and the Stockmann Foundation. We acknowledge the computational resources provided by the ELIXIR node, hosted at the CSC–IT Center for Science, Finland, and funded by the Academy of Finland (#271642 and #263164), the Ministry of Education and Culture, Finland. This study makes use of data generated by the UK10K Consortium, derived from samples from ALSPAC and TwinsUK. A full list of the investigators who contributed to the generation of the data is available from www.UK10K.org. Funding for UK10K was provided by the Wellcome Trust under award WT091310.

## Author contributions

E.K, O. Kuismin, K.R., S.S., M.S., K.K.E., J.T., O. Kilpivaara, and L.A.A. designed the study. K.R., D.G.B., E.A.M.H., and M.A. conducted the experiments. M.A., A.K., M.T., and P.V. acquired the sequencing data. E.K., H.R., R.K., A.T., and O. Kilpivaara analyzed the sequencing data. A.A., R.L., and E.S. contributed to the data analysis. O. Kuismin, J.N., J.J., K.K., and M.S. performed clinical evaluation of the study subjects. K.F.

diagnosed the tumor biopsies. O. Kuismin, M.M., A.P., M.K., O.P., M.H., R.C.S., and K. A. contributed to the acquisition of patient samples. E.K., O. Kuismin, K.R., J.K., T.T., M. S., K.K.E., O. Kilpivaara, and L.A.A. wrote the manuscript. E.K., K.R., A.K., and E.M. prepared the figures. E.K., O. Kuismin, K.R., J.K., M.A., A.K., T.T., J.N., J.J., K.K., M.S., K. K.E., J.T., O. Kilpivaara, and L.A.A. edited the manuscript. All authors approved the final manuscript.

## Additional information

**Competing interests:** The authors declare no competing interests.

Eevi Kaasinen[1,2,3,4], Outi Kuismin[5,6,7], Kristiina Rajamäki[1,2,8], Heikki Ristolainen[1,2], Mervi Aavikko[1,2], Johanna Kondelin[1,2], Silva Saarinen[1,2], Davide G. Berta[1,2], Riku Katainen[1,2], Elina A.M. Hirvonen[1,2], Auli Karhu[1,2], Aurora Taira[1,2], Tomas Tanskanen[1,2], Amjad Alkodsi[2], Minna Taipale[3,4], Ekaterina Morgunova[3,4], Kaarle Franssila[9], Rainer Lehtonen[2], Markus Mäkinen[10], Kristiina Aittomäki[11], Aarno Palotie[7,12,13], Mitja I. Kurki[12], Olli Pietiläinen[13], Morgane Hilpert[14], Elmo Saarentaus[7], Jaakko Niinimäki[15,16], Juhani Junttila[17], Kari Kaikkonen[17], Pia Vahteristo[1,2], Radek C. Skoda[14], Mikko R.J. Seppänen[18,19], Kari K. Eklund[8,20,21], Jussi Taipale[2,3,4], Outi Kilpivaara[1,2] & Lauri A. Aaltonen[1,2,3]

[1]Department of Medical and Clinical Genetics, University of Helsinki, FI-00014 Helsinki, Finland. [2]Genome-Scale Biology, Research Programs Unit, University of Helsinki, FI-00014 Helsinki, Finland. [3]Department of Biosciences and Nutrition, Karolinska Institutet, SE 171 77 Stockholm, Sweden. [4]Department of Medical Biochemistry and Biophysics, Karolinska Institutet, SE 171 77 Stockholm, Sweden. [5]Department of Clinical Genetics, Oulu University Hospital, FI-90029 Oulu, Finland. [6]PEDEGO Research Unit, Medical Research Center Oulu, Oulu University Hospital and University of Oulu, FI-90014 Oulu, Finland. [7]Institute for Molecular Medicine Finland (FIMM), HiLIFE, University of Helsinki, FI-00014 Helsinki, Finland. [8]Clinicum, University of Helsinki, FI-00014 Helsinki, Finland. [9]HUSLAB, Helsinki University Hospital, FI-00029 Helsinki, Finland. [10]Cancer and Translational Medicine Research Unit, University of Oulu, FI-90014 Oulu, Finland. [11]Department of Clinical Genetics, Helsinki University Hospital, FI-00029 Helsinki, Finland. [12]Analytic and Translational Genetics Unit, Department of Medicine, Department of Neurology and Department of Psychiatry, Massachusetts General Hospital, Boston 02114 MA, USA. [13]The Stanley Center for Psychiatric Research and Program in Medical and Population Genetics, The Broad Institute of MIT and Harvard, Cambridge 02142 MA, USA. [14]Department of Biomedicine, Experimental Hematology, University Hospital Basel and University of Basel, Basel CH-4031, Switzerland. [15]Medical Research Center Oulu, Oulu University Hospital and University of Oulu, FI-90014 Oulu, Finland. [16]Research Unit of Medical Imaging, Physics and Technology, Faculty of Medicine, University of Oulu, FI-90014 Oulu, Finland. [17]Research Unit of Internal Medicine, Medical Research Center Oulu, Oulu University Hospital and University of Oulu, FI-90014 Oulu, Finland. [18]Adult Immunodeficiency Unit, Infectious Diseases, Inflammation Center, University of Helsinki and Helsinki University Hospital, FI-00029 Helsinki, Finland. [19]Rare Diseases Center, Children's Hospital, University of Helsinki and Helsinki University Hospital, FI-00029 Helsinki, Finland. [20]Department of Rheumatology, Helsinki University Hospital, FI-00029 Helsinki, Finland. [21]ORTON Orthopaedic Hospital, FI-00280 Helsinki, Finland. These authors contributed equally: Eevi Kaasinen, Outi Kuismin, Kristiina Rajamäki.

