## [Peer Review File · Nature Communications]

Reviewers' Comments:

Reviewer #1:

Remarks to the Author:

The authors have answered most of the reviewers' comments, and indeed, it seems that they have identified a family with a germline TET2 haploinsufficiency with effects on DNA methylation.

The lack of correlation with any clear phenotype reduces the novelty and the ability to fully understand the significance of the results.

The fact the mutation does not segregate the Hodgkin Disease (HD) is actually predictable as TET2 mutations were not found among HD cases, but can be found in DLBCL. TET2 mutations occur also in AITL, AML, MPNs, MDS and Mast cell neoplasms. I agree with the authors that the penetrance can be very low in these cases, however understanding why it is low would be a finding of great importance. For example in a similar way the Tatton brown cases which rarely develop AML, however 3 cases are presented with young age AML 2 with the most common mutation (R882H Am J Med Genet A. 2017 Jan;173(1):250-253); J Med Genet. 2017 Dec;54(12):805-808. doi: 10.1136 and the other again in a hotspot at position 735 (Wellcome Open Res. 2018 Apr 23;3:46. doi: 10.12688). Also it is known that while DNMT3a is the most common mutation among individuals with clonal hematopoiesis the mutation position distribution is shifted in the AML cases towards more recurrent hot spots. This suggests that not all DNMT3a mutations can equally contribute to the risk of AML. A similar scenario can be true for TET2, however the current study shed no light on this. Most TET2 mutations in all Hem malignancies are frameshift and nonsense suggesting a haploinsufficiency in a similar way to the results of the current study, however the lack of a clear phenotype suggests that a major point is still unclear and the current study does not resolve it.

Regarding the phenotype: the TET2 mutation does segregate with HD in the family (that is, all individuals with HD carry the mutation), and yes the referee is right that not unexpectedly, also disease-free younger carriers are seen. We also agree, as explained in our first-round rebuttal, that from the limited number of individuals, firm conclusions regarding the full spectrum of phenotypes associated with TET2 mutation should not be drawn; time will tell as more cases are reported and selection bias attenuates. We also note that the index of the family originally discovered by Dr Radek Skoda (individuals with TET2del4 in our manuscript) does have a neoplasia of the myeloid lineage.

To thoroughly address this question, unbiased identification of TET2 germline mutation positive individuals from massive population-based cohorts will be required.

Regarding the functional effect of our mutations we note that all three are protein-truncating mutations that map safely among the dozens if not hundreds of similar (somatic) truncating TET2 mutations that have been reported causing hematological neoplasia.

The fact that no CVD phenotype nor monocyte hyperactivity could be observed in the current study could be related to the fact that the original studies by Fuster et.al and Jaiswal et.al used a KO model of TET2 in mice with hypercholesterolemia. Both the dosage differences and the exposure to high LDL levels could have accounted to the differences between the current study and previous ones. Jaiswal et.al identified high cytokine levels in monocytes in vitro after exposure to LDL.

That our work has been done in human rather than mouse context should indeed be considered as one important strength of our paper. We would also like to present the following considerations:

First, in the studies by Fuster et al. and Jaiswal et al., the cultured mouse macrophages used for probing cytokine and chemokine responses were not derived from hypercholesterolemic mice, but from Tet2^{-/-} and Tet2^{+/+} mouse strains fed with a normal diet; the Ldlr^{-/-} mice fed with a high cholesterol diet were only used in bone marrow transplantation experiments. Thus, the observed differences in IL-1beta response were not dependent on in vivo preconditioning in a hypercholesterolemic environment.

Second, Jaiswal et al. found increased native human LDL-induced gene expression of IL-1beta in Tet2^{-/-} mouse macrophages (Figure S7, A-B), yet the LDL-induced secretion of mature IL-1beta protein was negligible (Figure S7, C), thus offering no insight regarding the biologically most relevant IL-1beta output. Similarly, native and even oxidized LDL alone are very weak triggers of IL-1beta secretion in cultured human macrophages (Lehti, S. et al. Am J Pathol. 2018 Feb;188(2):525-538), and we thus chose – as did Fuster et al. in a combination treatment – to utilize the more established and robust atherosclerosis-associated inflammasome trigger cholesterol crystals. Notably, we have shown that cultured human macrophages also extract cholesterol from cholesterol crystals and accumulate it intracellularly as cholesteryl esters (Rajamäki, K et al. PLoS One. 2010 Jul 23;5(7):e11765.), thus driving the same key proatherogenic pathway of lipid-filled macrophage “foam cell” formation as LDL, along with the associated modifications to cellular signaling.

Third, Fuster et al. mainly employed LPS + IFN γ \pm ATP treatment to demonstrate elevated IL-1beta expression and secretion in cultured Tet2^{-/-} mouse macrophages, independent of the presence of LDL. In the current study, we have faithfully repeated this treatment combination with only minor modifications in human monocytes and macrophages.

In summary, although some differences in experimental setup are inevitable, the current manuscript does offer a thorough set of in vitro experiments for comparison to these mouse studies. Finally, Jaiswal et al. found significantly elevated LPS-induced IL-1beta secretion in Tet2^{+/+} + wild-type macrophages compared to Tet2^{-/-} (Figure S7, C; note also log scale), in stark contrast to results by Fuster et al. Thus, contradictory findings regarding IL-1beta were made already in the Tet2^{-/-} mouse macrophage studies, calling for caution and further studies on this proposed inflammatory mechanism.

Accordingly clear conclusions cannot be drawn on the phenotypic consequences of the mutation described besides the effect on DNA methylation, such a negative result could be due to many epistatic factors or environmental factors which were not identified or discussed in the current study.

This comment is somewhat difficult to follow. The key consequence of TET2 mutations in hematological neoplasia is epigenetic instability, and all four expert reviews including Ref #1 agree that our setting clearly recapitulates this key finding. However and importantly, we cannot see in humans the inflammatory phenotype, which based on the animal studies inherently results from the epigenetic instability state of the TET2 deficient mouse. This conclusion is also clear and of great interest for the community.

If the authors can come up with a clear explanation and mechanism why this family has no phenotypes which were reported in association with TET2 haploinsufficiency the manuscript will be at much better position.

See above, one of our mutation carriers does have a neoplasia of the myeloid lineage. Unbiased identification of TET2 germline mutation positive individuals from massive population-based cohorts will be required to fully elucidate the phenotypes associated with TET2 germline mutations.

Reviewer #2:

Remarks to the Author:

The authors have addressed my previous comments

We are glad to see that our extensive further work has satisfied the Reviewer.

Reviewer #3:

Remarks to the Author:

The authors made a good job to provide a lot of new data. Below are my comments

Regarding the rebuttal letter:

"Hydroxymethylation is not a stable modification on DNA and thus, we feel that creating hydroxymethylation data from the available sample materials (peripheral blood and lymphoblastoid cells) from the TET2-mutated individuals does not have the potential to reveal further significant differences associated with TET2 loss. The effects related to TET2 loss are likely most relevant during development. Unlike hydroxymethylation differences, DNA methylation differences that emerge during differentiation remain in the progeny, and thus could be observed in our analyses of the available sample materials."

Comment: The present reviewer would disagree with the blunt claim that "hydroxymethylation is not a stable modification on DNA". In a given cell type and in response to a specific stimulus, for example, it is sufficiently stable for analyses, as reported. This point could however be confirmed (or infirmed) by ChIP-QPCR with anti-hmC, at differentially H3K27acetylated in from immortalized lymphoblastoid cell lines. In addition it is not absolutely established that methylation differences are uniquely a scar of differentiation and not also selected later, during cellular proliferation.

We agree that hydroxymethylation is sufficiently stable for analysis in a given cell type and in response to stimulus. We should have been more precise in our response and say that hydroxymethylation is not a stable modification on DNA through cell divisions.

"Regarding the second point, we looked into FOXP3 expression in blood of cancer-free TET2delA carriers (n=3) and wild-type individuals (n=3), but very few cells in blood express the gene; see figure below."

Comment: This is trivial that because of the low depth of this single cell RNA-seq approach, a given gene is detected only in a fraction a cells. Signature expression or predicted activity of a transcription factor should be searched for, has the author report for TBX21 and EOMES.

FOXP3 co-expressed target genes cannot be calculated as was done for TBX21 and EOMES, for example, because there were only 19/6664 cells that displayed FOXP3 expression in the six cancer-free individuals in total. A gene whose reads originate from very few cells can correlate with another gene by coincidence – hence, we studied the activity of every TF that was detected in at least 1% of the cells. Statistical significance for differential activity was calculated for 105 TFs that had enough expression and binding motifs available.

Two carriers have some immune-related diseases: rheumatoid arthritis, celiac disease. Any interesting details in there?

Nothing striking. This comment relates to that of Referee #1, and the question what is the full spectrum of phenotypes associated with TET2 germline mutations. To thoroughly address this question, unbiased identification of TET2 germline mutation positive individuals from massive population-based cohorts will be required.

Some of the negative observations, such as the lack of mutations signature in the genomic sequences (whole genome sequences, obviously?), should be mentioned somewhere.

Mutation signatures were from deep exome sequences. We can report the negative observations but the manuscript is already now quite extensive and long; Editor please advise.

Abstract:

"58 Compatible with this, the regions displayed reduced methylation in DNMT3A

59 germline mutation carriers, likely due to TET2-mediated oxidation. "

Comment: This is unclear: The present reviewer understood that DNMT3A mutations would inactivate Methyl deposit, so be upstream of TET2-mediated oxidation

We agree that the sentence was not clear enough; we have modified the sentence as follows "Compatible with this, the regions [with binding sequences of master transcription factors] displayed reduced methylation relative to all open chromatin regions in four DNMT3A germline mutation carriers, likely due to TET2-mediated oxidation" In other words, TET2 is recruited particularly to these sites by the TFs, thus the methylation pattern we see.

"59 Our findings provide new

60 insight into the putative role of TET2 in atherosclerosis, and the interplay between epigenetic 61 modulators and transcription factor activity in hematological neoplasia."

Rather: The findings do not confirm "the putative role of TET2 in atherosclerosis" at least in the setting of heterozygous truncating TET2 germline mutation.

We have modified the sentence as follows "Our findings provide new insight into the interplay between epigenetic modulators and transcription factor activity in hematological neoplasia, but do not confirm the putative role of TET2 in atherosclerosis".

"132 analyses (Supplementary Table 3). In addition, we had the opportunity to study innate immune responses

133 in blood monocytes isolated from two carriers of a heterozygous four base pair deletion in TET2"

Page 7

"144 points and because the chromatin annotation of affected regions after TET2 loss is likely identical

145 in different cell types."

Comment: This statement is not fully correct. Many people and EU are spending a lot of effort and money to map epigenetics marks and annotate genome with respect to specific cell types. TET2 is at least partially recruited to chromatin by lineage-specific transcription factors.

Regarding page 6 we could not identify a question, please clarify.

Regarding page 7, maybe some misunderstanding here. We are in full agreement with the reviewer that TET2 is at least partially recruited to chromatin by lineage-specific transcription factors. See e.g. our conclusions in the manuscript in the second and third paragraph in the discussion. Although genomic regions where TET2 is recruited are different in different cell types, their chromatin annotation e.g. active chromatin marked by H3K27ac is likely the same.

146 TET2delA carriers displayed a significantly higher degree of overall hypermethylation (two-sided

147 Wilcoxon rank sum test, p-value 0.003) and decreased hypomethylation (two-sided Wilcoxon rank

148 sum test, p-value 0.01), compatible with demethylation deficiency (Fig. 2a, Supplementary Table 149 5)."

see also page 16 lanes 355-356

Comment: Most of the CpGs of the genome are in repeated regions. Are the authors stating that these regions are hypermethylated ? Or are they refereering only to single copy genomic regions?

We emphasize that the methylation values are extracted from whole-genome and targeted bisulfite sequencing of ≥ 100 bp paired-end reads, and for each read pair a unique best alignment in the genome is determined. Thus, the results should not be strongly biased by repetitive genomic regions.

As suggested by the reviewer, we checked whether hypermethylated CpGs are enriched in repeated regions (see new Supplementary Fig. 5a). Hypermethylated CpGs locate mostly at CpG islands but also to some degree at simple repeats of the genome.

"161 Hypermethylation

162 was more prominent than hypomethylation when WGBS data of the three cancer-free TET2delA

163 carriers and the TET2X carrier were compared to age-matched non-carriers (Fig. 2b-f)."

Comment: All patients had a comparable blood composition at sampling, hadn't they?

Blood composition of TET2delA and TET2X carriers at the time of sampling for WGBS is presented in Supplementary Tables 9 and 10. Also, the fraction of major blood cell types in TET2delA and age-matched non-carriers are shown to be similar by single-cell profiling at the time of sampling for WGBS (see Supplementary Fig. 14). Blood composition of age-matched NFID cohort controls (Id2 and Id3) was not studied. We do agree that blood composition is a potential confounding factor and for this reason, we have not made strong claims about specific differentially methylated regions associated with TET2 loss. However, the direction of methylation effect after TET2 loss is consistently towards hypermethylation at active enhancer regions despite minor differences in blood composition between mutation carriers and non-carriers.

Page 12

"266 No somatic TET2 mutations were detected, 267 suggesting that complete TET2 loss provided little selective advantage."

Comment: Based on exome data of two samples? This is an overstatement. For example, can the authors exclude the presence of a minor (5-15%) clone carrying a deletion of an exon of the wild type copy of the gene? or a chromosomal translocation disrupting the wild type copy of TET2? or a deletion of the TET2 promoter?

We deep exome sequenced five mutation carriers, each having thousands of hematopoietic stem cells, and did not encounter a single clone with evidence of two hits. Thus, we find the expression "suggesting" appropriate. For example, complete loss of APC gene in colonic epithelium is highly selected for and indeed in APC germline mutation carriers one sees hundreds or thousands of double APC loss neoplastic clones emerging relatively early in life, through inactivation of the normal allele by somatic point mutations or allelic loss. We do acknowledge that it is not possible to be sure that in no cell in our carriers there is a second hit through allelic loss, and that is why we have used the word "suggesting". However, if even after this explanation this expression is deemed too strong we can revise and say instead for example "No somatic TET2 mutations were detected, compatible with data in clonal hematopoiesis where only one TET2 mutant allele is typically seen within a clone". We would prefer the original wording but will change if advised by the Ref / editors. See also the considerations below, responding to Ref #5.

"Page 15-16, lanes 362-369. "

The present manuscript provides experimental data in two examples. There are however several publications in the literature showing that.

We can add more references but the current number of references is already over the limit for publication in Nature Communications; Editor please advise.

Reviewer #4:

Remarks to the Author:

My comments have been addressed and there is a significant amount of new data in the manuscript that confirms the major findings.

We are glad to see that our extensive further work has satisfied also this Reviewer.

Reviewer #5:

Remarks to the Author:

In this manuscript, Kaasinen et al report on a family with a Tet2 haploinsufficiency. The authors analyzed the blood methylome, immune cells by single-cell RNA-sequencing, inflammatory responses, and atherosclerosis. There is a lot of very interesting information in this manuscript. The authors, however, are most provocative when they suggest that Tet2 haploinsufficiency does not have any effects on the inflammatory capacity of monocytes/macrophages (specifically IL-1b) and has no effect on atherosclerosis. This is provocative as several high profile papers have shown in the mouse that Tet2 deficiency in macrophages promotes IL-1 production and aggravates atherosclerosis (notably, Fuster et al, Science, 2017). The suggestion here is that perhaps the mouse work does not recapitulate human disease. While this may be so, the manuscript does not make a strong case to seriously challenge the mouse studies.

First, the mouse studies used Tet2-deficient mice while here the authors use cell from humans with Tet2 haploinsufficiency. Moreover, the authors silence Tet2, but the silencing is incomplete. While the haploinsufficiency may be sufficient to uncover the many other phenotypes, it remains to be determined whether the residual Tet2 is enough to affect the inflammatory cascade. Therefore, the highly provocative statement is not supported by the data.

Second, the authors rely on a handful of individuals to make a case that Tet2 insufficiency does not aggravate atherosclerosis. This is entirely anecdotal and is not supported by the data. While it may be interesting to know how someone died, conclusions as to their atherosclerotic disease cannot be made.

Thus, while the authors may wish to speculate whether Tet2 has an effect on the inflammasome and atherosclerosis, this part of the manuscript is quite weak and any strong conclusions are premature.

We respectfully disagree with both points raised by the referee.

First point: The reviewer is understandably very negative because apparently s/he has fundamentally misunderstood the setting. S/he is judging whether our data would appropriately test whether the mouse work using TET2 k-o cells by e.g. Fuster et al. (Science 2017) and Jaiswal et al.

(NEJM 2017), mechanistically linking TET2 loss to atherosclerosis and raising enormous attention, is correct. We fully agree that our experimental setting is different, and indeed the human TET2-/+ heterozygous cells are a suboptimal model for TET2-/- cells derived from an entirely different species. Of note, we do not believe that the mouse work is necessarily wrong, and we do not make such a claim in the paper.

However, the correct question for human health of course is whether the atherogenic mechanism shown in mice by Fuster et al. operates in individuals with clonal hematopoiesis carrying a blood cell subclone with a somatic TET2 mutation. These mutations are common in the population - thus the

great interest and importance - and they are typically heterozygous. Perhaps Referee #5 simply does not know this. This would be understandable. If one studies the many papers on the subject, the heterozygous status of the TET2 mutations found in clonal hematopoiesis is not often very clearly stated. Sometimes you need to examine the supplement to find this information, see for example Jaiswal et al. *New England Journal of Medicine* 371(26):2488-98, 2014. In that paper one needs to find the Supplementary Table S3, copy the patient Ids to excel to be able to sort the information, and sort by Id number to see that all 73 TET2 mutations were heterozygous (all 73 mutations were detected in different patients; no patient contributed two mutations). One could of course speculate that might a second hit have occurred through allelic loss and that indeed is a valid point. This has rarely been examined but to this end we paste below the abstract from Smith et al *Blood* 2010; this is a rare example of a paper which also addresses the frequency of copy number loss at TET2; it occurs sometimes but much more often not. To our knowledge, in no publication premalignant clones are shown to be predominantly TET2^{-/-}. Should the Reviewer wish to adhere to his/her view we would appreciate if s/he could provide some factual basis for Tet2^{-/-} mouse cells being a more appropriate model as compared to TET2^{-/+} human cells to study effects of common CH in human atherosclerosis.

To study the human clinical relevance of the mechanisms described by Fuster and others one thus needs human TET2^{-/+} cells (or eg siRNA TET2 silencing to 50%).

This is exactly what we have done. The mechanism published in *Science* could not be reproduced, in the setting relevant to humans. All four expert reviewers above agree that we show the TET2 haploinsufficient individuals to display the expected epigenetically unstable state, faithfully reproducing the methylation phenomena seen in clonal hematopoiesis. None of them express any doubts. This "model" is much more relevant to the clinical research question than the Tet2 null mice.

Abstract from Smith et al *Blood* 2010

Next-generation sequencing of the TET2 gene in 355 MDS and CMML patients reveals low-abundance mutant clones with early origins, but indicates no definite prognostic value

Alexander E. Smith, Azim M. Mohamedali, Austin Kulasekararaj, ZiYi Lim, Joop Gäken, Nicholas C. Lea, Bartłomiej Przychodzen, Syed A. Mian, Erick E. Nasser, Claire Shooter, Nigel

B. Westwood, Corinna Strupp, Norbert Gattermann, Jaroslaw

P. Maciejewski, Ulrich Germing and Ghulam J. Mufti *Blood* 2010 116:3923-3932; doi: <https://doi.org/10.1182/blood-2010-03-274704>

Abstract

Mutations in the TET2 gene are frequent in myeloid disease, although their biologic and prognostic significance remains unclear. We analyzed 355

patients with myelodysplastic syndromes using “next-generation” sequencing for TET2 aberrations, 91 of whom were also subjected to single-

nucleotide polymorphism 6.0 array karyotyping. Seventy-

one TET2 mutations, with a relative mutation abundance (RMA) $\geq 10\%$, were

identified in 39 of 320 (12%) myelodysplastic syndrome and 16 of 35 (46%) chronic myelomonocytic leukemia patients ($P < .001$). Interestingly, 4 patients

had multiple mutations likely to exist as independent clones or on alternate

alleles, suggestive of clonal evolution. “Deeper” sequencing of 96 patient

samples identified 4 additional mutations (RMA, 3%-6.3%).

Importantly, TET2 mutant clones were also found in T cells, in addition to

CD34+ and total bone marrow cells (23.5%, 38.5%, and 43% RMA, respectively). Only 20% of the TET2-mutated patients showed loss of

heterozygosity at the TET2 locus. There was no difference in the frequency of

genome-wide aberrations, TET2 expression, or the JAK2V617F 46/1

haplotype between TET2-mutated and nonmutated patients. There was no significant prognostic association between TET2 mutations and World Health

Organization subtypes, International Prognostic Scoring System score,

cytogenetic status, or transformation to acute myeloid leukemia. On multivariate analysis, age (> 50 years) was associated with a higher incidence of TET2 mutation ($P = .02$).

of TET2 mutation ($P = .02$).

Second point: A key question here is, does heterozygous TET2 loss that occurs late in life through a somatic mutation potentially lead to a clinically significant atherosclerosis promoting state. Such individuals are exposed to a - often minor - TET2 mutant subclone, and the number of exposure years is limited. Our mutation carriers have been exposed to effects of TET2 deficiency since embryonic life, and the number of mutant cells vastly exceeds the ones seen in context of CH. Thus, it is not uninteresting to see that no striking effects are seen. This observation is also compatible with our (negative) data on inflammasome activation under TET2 haploinsufficiency. Surely these novel observations should be shared by the community, in particular as the epidemiological data associating CH and cardiovascular disease is not free of problems. The epidemiological positive association between CH and cardiovascular disease seen in some - but not all - studies may represent simply the fact that different individuals age at different pace. With the increasing biological age comes increased inflammation and consequently - as a bystander - associations between inflammation-linked diseases such as cancer and cardiovascular disease. Importantly, the TET2 germline mutant individuals provide an independent angle to the matter.

We do agree with the message of the reviewer that a good scientific paper presents findings with appropriate caution, and we have modified the discussion even further to that direction. For details see the discussion of our paper.

Comment by Reviewer #3:

As most of the genomic investigation, the analyses are restricted to single copy regions ("**a unique best alignment**"), so the conclusions do not apply for repeated regions, which consist most of the genome methylation. My understanding is that global genome cytosine methylation is not markedly different between Tet2-KO and WT samples, although unique copy regions are hypermethylated. Please be precise to avoid misleading the readers

Our response:

To avoid misleading the readers, we have now modified the text to include: "While repetitive regions of the genome cannot be evaluated with short reads, the detected hypermethylated CpG sites located mostly at CpG islands of the genome (Supplementary Fig. 5a)."